# High-resolution kinetic characterization of the RIG-I-signaling pathway and the antiviral response

Sandy S Burkart[1,2,]*, Darius Schweinoch[3,]*, Jamie Frankish[1,2], Carola Sparn[1,2], Sandra Wüst[1], Christian Urban[4], Marta Merlo[1,2], Vladimir G Magalhães[1], Antonio Piras[4], Andreas Pichlmair[4,5], Joschka Willemsen[1], Lars Kaderali[3], Marco Binder[1]

**RIG-I recognizes viral dsRNA and activates a cell-autonomous antiviral response. Upon stimulation, it triggers a signaling cascade leading to the production of type I and III IFNs. IFNs are secreted and signal to elicit the expression of IFN-stimulated genes, establishing an antiviral state of the cell. The topology of this pathway has been studied intensively, however, its exact dynamics are less understood. Here, we employed electro-poration to synchronously activate RIG-I, enabling us to characterize cell-intrinsic innate immune signaling at a high temporal resolution. Employing IFNAR1/IFNLR-deficient cells, we could differentiate primary RIG-I signaling from secondary signaling downstream of the IFN receptors. Based on these data, we developed a comprehensive mathematical model capable of simulating signaling downstream of dsRNA recognition by RIG-I and the feedback and signal amplification by IFN. We further investigated the impact of viral antagonists on signaling dynamics. Our work provides a comprehensive insight into the signaling events that occur early upon virus infection and opens new avenues to study and disentangle the complexity of the host–virus interface.**

## Introduction

Recognition of pathogen-associated molecular patterns (PAMPs) by a variety of cell-surface and intracellular pattern-recognition receptors (PRRs), such as retinoic acid-inducible gene-I (RIG-I)-like receptors (RLRs), can trigger signaling cascades leading to the expression of cytokines, for example, IFNs, and IFN-stimulated genes (ISGs). Briefly, upon viral dsRNA recognition, RIG-I initiates a signaling cascade through mitochondrial antiviral-signaling

protein (MAVS), leading to the activation and nuclear translocation of transcription factors such as IFN-regulatory factor 3 (IRF3) and NF-κB. Activation of these transcription factors results in the production and secretion of mostly type I and III IFNs. Auto- and paracrine IFN signaling through the corresponding IFN receptors (IFNR) can further induce multiple downstream signaling pathways. The most important signaling axis is the JAK/STAT pathway, leading to the formation of a transcription factor composed of the STAT of transcription 1 (STAT1), STAT2, and IRF9, in conjunction known as the ISGF3 complex. Upon activation, this complex binds to IFN-stimulated response elements within gene promoters, further leading to the induction of a large variety of ISGs, which exert numerous antiviral functions (Schoggins et al, 2011; Schoggins, 2019). In addition, upon activation by the only type II IFN, known as IFN-γ, STAT1 can form homodimers that bind to IFN-γ–activated sequence elements in promoter regions of ISGs. Thus, the expression of both, IFNs and ISGs, ultimately establishes an antiviral state of a cell (Nakhaei et al, 2009; Onomoto et al, 2010; Yoneyama et al, 2015; Chow & Loo, 2018). Considering the important impact of the IFN system on the outcome of viral infection, viruses have evolved strategies to evade innate immune recognition and defense (Haller et al, 2006; Beachboard & Horner, 2016). One strategy is to omit recognition by PRRs in the first place, with the hepatitis B virus being a prime example (Wieland & Chisari, 2005; Mutz et al, 2018). However, this appears excessively difficult to achieve, and most viruses have evolved antagonists of host–cellular antiviral pathways (García-Sastre, 2004, 2017; Wieland & Chisari, 2005). These virus-encoded factors target different cellular processes (Pichlmair et al, 2012). For instance, the classical swine fever virus (CSFV) protein N^pro was identified as an antagonist of RIG-I signaling by targeting IRF3 for degradation (Bauhofer et al, 2007; Seago et al, 2007). Similarly, the hepatitis C virus (HCV) protease NS3/4A targets MAVS for cleavage and release from the mitochondrial membrane,

---

[1]Research Group "Dynamics of Early Viral Infection and the Innate Antiviral Response", Division Virus-Associated Carcinogenesis (F170), German Cancer Research Center, Heidelberg, Germany    [2]Faculty of Biosciences, Heidelberg University, Heidelberg, Germany    [3]Institute of Bioinformatics & Center for Functional Genomics of Microbes, University Medicine Greifswald, Greifswald, Germany    [4]Technical University of Munich, School of Medicine, Institute of Virology, Munich, Germany    [5]German Center for Infection Research (DZIF), Munich Partner Site, Munich, Germany

Correspondence: m.binder@dkfz.de; lars.kaderali@uni-greifswald.de
Jamie Frankish's present address is Avectas, Maynooth, Ireland
Joschka Willemsen's present address is Novartis Institutes for BioMedical Research, Novartis Campus, Basel, Switzerland
*Sandy S Burkart and Darius Schweinoch contributed equally to this work

---

rendering RLR/IRF3-signaling dysfunctional (Bartenschlager et al, 1993; Eckart et al, 1993; Li et al, 2005). Viral antagonists have been studied extensively, and many have been characterized in detail. Nonetheless, most studies investigating such viral antagonists are limited to robust overexpression and end-point determination of the degree of inhibition of the host response. However, in an actual infection, active concentrations of the antagonists are gradually increasing as the virus replicates and expresses its genes. Moreover, depending on the target of inhibition, the resulting effect may either delay the mounting of an antiviral response or lead to an overall lower amplitude of the response (reviewed in Haller et al [2006]; Beachboard & Horner [2016]; Kikkert [2020]). Therefore, to comprehensively understand viral immune evasion, it is crucial to investigate the dynamics of the cellular antiviral response and the impact of antagonists on these dynamics ("dynamics" referring to the combination of kinetics and amplitude). Particularly for viruses, this system is largely based on cytoplasmic sensors, such as the RIG-I–like receptors. Whereas simultaneous activation of cell surface receptors such as TNF receptor (Amarante-Mendes et al, 2018), IL-1 receptor (Dinarello, 2017), and some TLRs (El-Zayat et al, 2019) can easily be achieved by ligand addition to the cell culture media, this is particularly challenging for intracellular receptors such as RIG-I. Owing to this cytosolic localization of the receptors, instant stimulation of the pathway is challenging, hence virtually, all kinetic studies were based on actual virus infection or liposome-based transfection of virus-like RNA (Rand et al, 2012; Esser-Nobis et al, 2020; Thoresen et al, 2022). However, cellular uptake of the PAMP (i.e., viral RNA) in those cases depends on endocytic processes which introduce delays and large variability in the timing of the PAMP to be accessible to PRRs.

In this study, we kinetically characterized the dynamics of antiviral signaling triggered by RIG-I in human alveolar epithelial (A549) cells and established a mathematical model which allows accurate dynamic simulation. We established an approach permitting the virtually instant stimulation of A549 cells with virus-like dsRNA, leading to a synchronous onset of signaling across all cells and, thus, allowing for a finer resolution of pathway kinetics. With this system, we analyzed phosphorylation and expression of critical proteins in the RIG-I signaling cascade in a fine-grained time course using live-cell imaging, quantitative Western blotting, and qRT–PCR. We used these time-resolved data to establish and calibrate a mathematical model that reproduces the kinetics of key signaling events within the core RIG-I pathway. We combined our RIG-I core model with a previously published and validated model of IFN signaling (Maiwald et al, 2010), ultimately generating a comprehensive model of the cell-intrinsic antiviral system. Furthermore, using viral antagonists with well-established mechanisms of action, we characterized their impact on RIG-I and IFN-signaling dynamics and compared experimental data with model simulations. We present the most comprehensive, data-based mathematical model of the cell-intrinsic antiviral defense system, permitting simulation and analysis of critical virus–host interactions early into infection.

# Results

## RIG-I signaling upon dsRNA recognition is highly deterministic

In previous studies, RLR signaling kinetics upon viral infection or liposome-based transfection resulted in a heterogeneous nuclear translocation of transcription factors, such as IRF3 or IRF7, or expression of the *IFNB1* gene (Rand et al, 2012; Doğanay et al, 2017). For instance, Rand and colleagues reported cell-intrinsic stochasticity in the activation of IRF7 and NF-κB upon virus infection of murine cells (Rand et al, 2012). To determine the dynamics of RLR signaling, we employed A549 cells stably co-expressing cytosolic IRF3-eGFP and nuclear H2B-mCherry and inspected nuclear translocation of IRF3 after RIG-I stimulation (Fig 1). As a ligand specific for RIG-I, 400-bp 5′ppp-dsRNA (Binder et al, 2011) was delivered to cells by liposome-based transfection or electroporation-based transfection. In electro-transfection, uptake of the RNA is largely restricted to the sub-second period of the electrical pulse, whereas RNA-laden liposomes ("lipoplexes") are slowly taken up by cells over the course of several hours. In contrast to the literature, we found RIG-I signaling to commence synchronously across cells and to be highly deterministic, that is, to lack signs of stochasticity or cell-to-cell variability, after electro-transfection of 5′ppp-dsRNA. Mock electro-transfection did not induce innate immune signaling (Fig S1A and B). Confocal microscopy analysis of IRF3-eGFP nuclear translocation kinetics (Fig 1A) demonstrated a clear difference between the methods of RLR ligand delivery. Whereas electro-transfected cells showed a rapid and synchronous IRF3 translocation, liposome-based transfection of 5′ppp-dsRNA resulted in a staggered (asynchronous) translocation in A549 cells (Fig 1A). It needs to be noted that ectopic expression of IRF3-eGFP increases total levels or IFR3 in these cells, which might slightly alter absolute activation and translocation dynamics, but is unlikely to affect electro-transfected and liposome-transfected cells differently. Using live-cell imaging and automated image analysis to quantitatively assess nuclear translocation after RIG-I stimulation, we again observed that liposome-based transfection of 5′ppp-dsRNA resulted in staggered IRF3 translocation with a constant increase of the fraction of cells exhibiting nuclear IRF3-eGFP over 5 h. In contrast, electro-transfection induced translocation synchronously across cells yielding a rapid increase of cells with nuclear IRF3-eGFP peaking 60 min after stimulation (Fig 1B). To determine whether the observed results were because of possible differences in the RNA amounts successfully entering the cytosol, we titrated the amount of 5′ppp-dsRNA. Indeed, activation kinetics were slightly affected and the maximum fraction of activated cells was significantly affected by decreasing dsRNA concentrations, but the qualitative characteristics remained consistent: independent of the dsRNA amount, electro-transfection led to substantially steeper activation kinetics (Fig 1C) as compared with liposome-based transfection (Fig 1D). These results corroborated our hypothesis that RNA-uptake kinetics of the chosen transfection modality plays the single-most decisive role in the previously described asynchronicity ("stochasticity") of RLR-driven IRF responses.

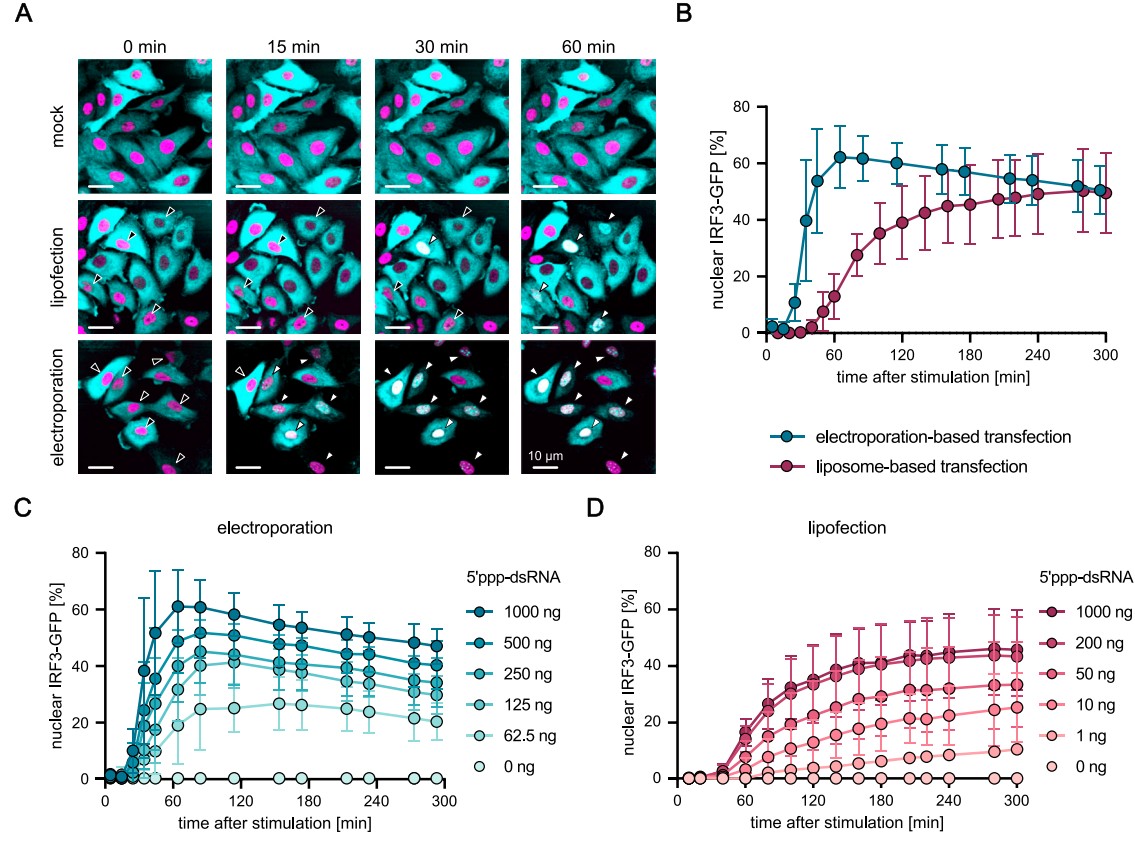

**Figure 1. Synchronous activation of the RIG-I signaling pathway upon dsRNA electro-transfection.**
**(A)** Confocal microscopy time course of A549 cells stably expressing cytoplasmic IRF3-eGFP (cyan) and the nuclear marker H2B-mCherry (magenta) upon different stimulation approaches. Cells were either mock-treated or stimulated with 1 μg in vitro-generated 5'ppp-dsRNA using a liposome-based transfection or in-well electro-transfection using the Lonza Nucleofector system. Black arrowheads indicate cytoplasmic IRF3-eGFP and white arrowheads indicate translocated, nuclear IRF3-eGFP. Scale bars measure 10 μm. **(B)** Quantification and comparison of IRF3-eGFP nuclear translocation upon stimulation with 1 μg 5'ppp-dsRNA using a liposome-based or electroporation-based transfection method. Nuclear translocation of IRF3-eGFP and co-localization with H2B-mCherry was analyzed over time using the Incucyte live-cell imaging system. Here, at least 500 and up to 2,500 individual cells were automatically evaluated for each time point in each condition using the Ilastik software. **(C)** Quantification and comparison of IRF3-eGFP nuclear translocation upon electroporation-based transfection of varying concentrations of 5'ppp-dsRNA. **(D)** Quantification and comparison of IRF3-eGFP nuclear translocation upon liposome-based transfection of varying concentrations of 5'ppp-dsRNA. **(A, B, C, D)** Graphs depict representative images (A), the mean ± SD of four (B) or three (C, D) biologically independent experiments, respectively.

## Synchronous RIG-I stimulation results in an immediate onset of RLR pathway signaling

Simultaneous activation of the RLR signaling pathway across cells by electro-transfection permitted us to kinetically characterize RIG-I signaling dynamics in more detail. We simultaneously stimulated A549 WT cells with 5'ppp-dsRNA and analyzed the activation (i.e., phosphorylation) status of RLR signaling pathway components or protein abundance (for IκB) within the IRF3– (Fig 2A) or NF–κB axis (Fig 2B) over time via Western blotting. Already 10 min after stimulation, activated and phosphorylated forms of the kinases TBK1 and IKKε (for IRF3) or IKKβ (for NF-κB) were detectable. The earlier onset of decay of the NF-κB inhibitor, IκBα, 5 min after transfection, which is dependent on IKKβ activation, argues for even earlier kinase activation (Fig 2B). IRF3 phosphorylation started shortly after TBK1 and IKKε activation (Fig 2A). Furthermore, whereas NF–κB and IKKβ kinase phosphorylation decreased concurrently (Fig 2B), IRF3 phosphorylation was surprisingly stable within the experimental time frame, despite pTBK1 and pIKKε levels declining

from 1 h onwards (Fig 2A). Consistent with the rapid onset of RIG-I signaling, early transcripts of the target genes *IFNB1*, *IFNL1*, and *IFIT1* (Fig 2C), and *CCL5* and *TNFAIP3* (Fig 2D) were detectable already at 45–60 min poststimulation by qRT–PCR.

## Dynamic RIG-I signaling model accurately reproduces activation kinetics of essential pathway components

Our acquired time-resolved, quantitative data on the activation kinetics of the canonical RIG-I pathway components provided an excellent description of the antiviral signaling dynamics in our experimental system. To generalize these observations and investigate kinetic characteristics such as rate-limiting steps in this pathway, we employed kinetic mathematical modeling. We developed a computational model using ordinary differential equations (ODE) representing main steps within the RIG-I signaling pathway based on regular mass-action kinetics. Our model comprises 19 pathway components ("species") distributed into two distinct compartments (cytoplasm and nucleus), 20 rate

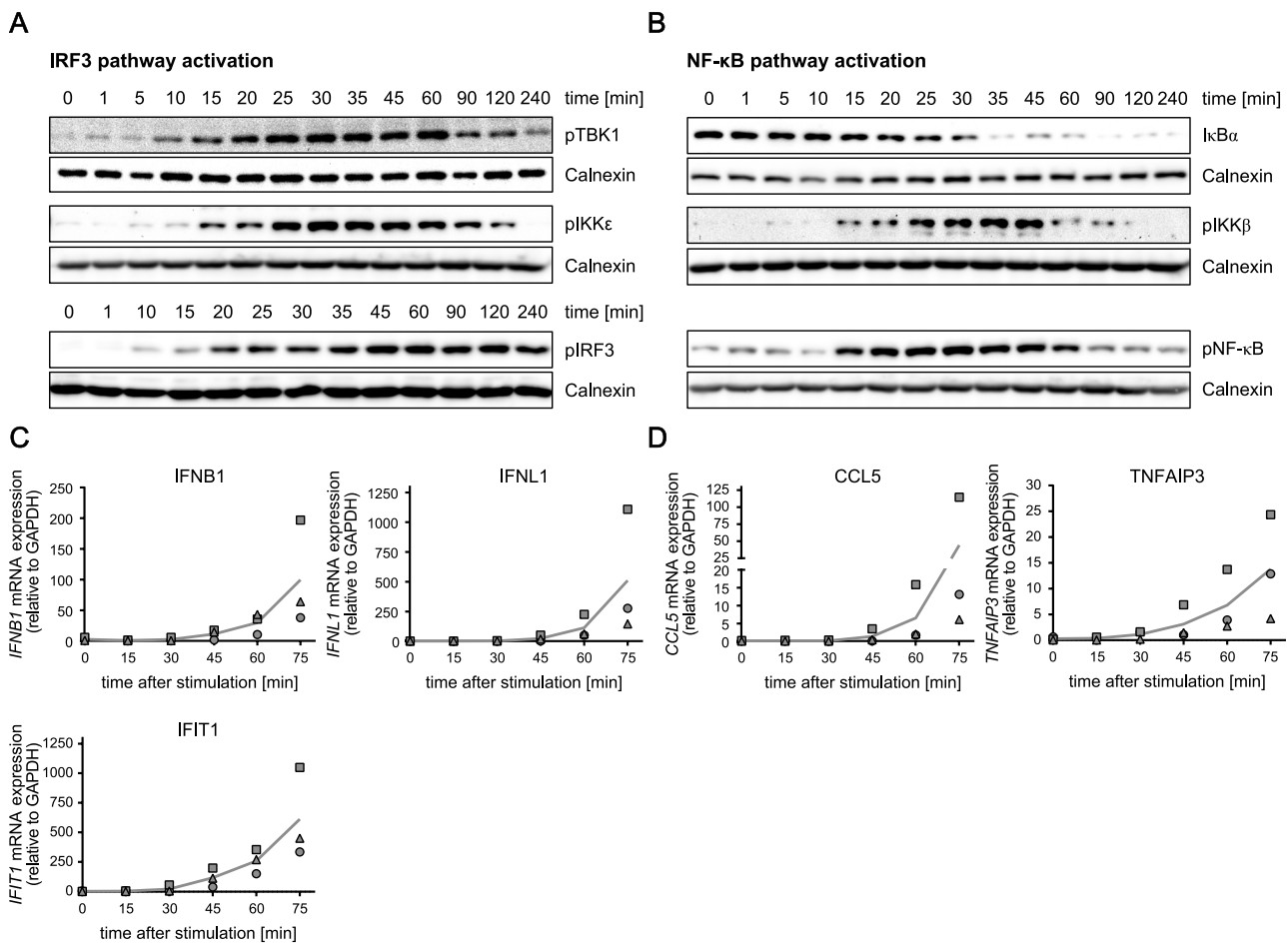

**Figure 2. Time-resolved activation of the early RIG-I signaling cascade.**
Synchronized stimulation of A549 WT cells with 220 ng 5'ppp-dsRNA by electro-transfection was used to kinetically characterize the dynamics of the RIG-I-signaling pathway. **(A, B)** Protein abundance and phosphorylation status of RLR signaling pathway components within the (A) IRF3 or (B) NF-κB axis were determined using Western blot analysis. **(C, D)** Early mRNA expression onset of (C) IRF3-dependent genes (*IFNB1*, *IFNL1*, and *IFIT1*) and (D) NF-κB-dependent genes (*CCL5* and *TNFAIP3*) after dsRNA electro-transfection was measured using qRT–PCR. Values were normalized to the housekeeping gene *GAPDH* using the $2^{-\Delta Ct}$ method. **(A, B, C, D)** Graphs depict representative blots (A, B) or the mean and individual, biological replicate values (C, D) of three biologically independent experiments.

constants, and some additional parameters (e.g., normalization factors) to account for initial conditions, experimental conditions, and the relative nature of the quantitative immunoblot data (Fig 3A). To determine absolute concentrations of all relevant proteins, we acquired quantitative full-proteome data of naïve A549 cells by label-free mass spectrometry. Kinetic rate constants that could not be fixed based on prior information were optimized during model fitting to relative protein quantities and phosphorylation intensities from immunoblotting (Fig 3B) and transcript levels from qRT–PCR data (Fig 3C) (see Supplemental Data 1 for modeling details). Our calibrated model was able to accurately and reliably capture the measured signaling dynamics (Figs 2 and 3B and C and Supplemental Data 1). To challenge the model with conditions not used for its training, we performed additional experiments in A549 cells with artificially lowered concentrations of IRF3. We used A549 IRF3 KO cells and restored IRF3 to ~5% of WT level by lentiviral transduction using a weak promoter (murine ROSA26) (Fig S2A). As expected, IRF3 KO

diminished *IFNB1* induction upon 5'ppp-dsRNA electro-transfection by more than 1,000-fold (Fig 3D). A549 ROSA26-IRF3 cells showed an intermediate phenotype with slower induction kinetics and >1 log₁₀-reduced transcript levels at 4 h poststimulation compared with WT cells (Fig 3D). Notably, this dampened induction kinetics was correctly captured by our model after adjusting only the IRF3 concentration, accurately predicting the qualitative changes, with even the quantitative predictions matching the actual measurements reasonably well (Fig 3E). For further model validation, we simulated the fraction of nuclear IRF3 upon stimulation with varying concentrations of 5'ppp-dsRNA (Fig S3), compared these model predictions with experimental data (Fig 1C), and observed a high agreement between experimentally obtained data and model simulation. This validation using independent data highlighted the usability of our ODE model not only to meaningfully fit measured data, but also to predict the outcome of signaling under perturbed conditions and, hence, to draw functional conclusions from in silico simulations.

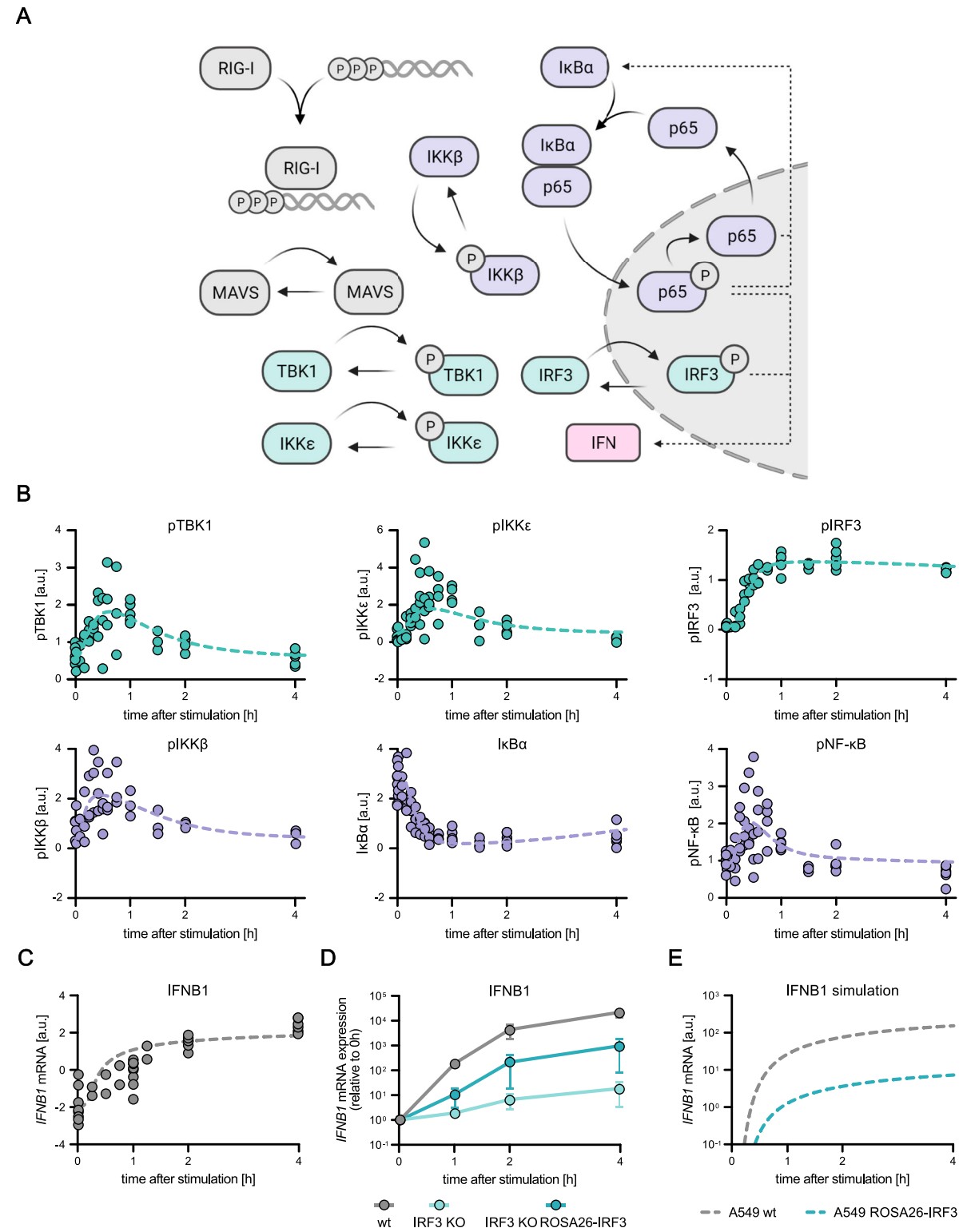

**Figure 3. Mathematical model of the core RIG-I pathway based on quantitative data.**
The previously obtained, time-resolved data were used to establish and calibrate a mathematical model which reproduces the kinetics of key signaling events within the core RIG-I pathway. **(A)** Schematic depiction of key components relevant to the establishment of a mathematical model of the core RIG-I signaling pathway. **(B, C)** Quantitative protein abundance, protein phosphorylation, and (C) mRNA data of A549 WT cells synchronously stimulated with 220 ng 5′ppp-dsRNA were used to set up and calibrate a dynamic mathematical model. **(B, C)** Dots represent quantitative data of four (B) or two (C) biologically independent experiments and lines represent average model fit. **(D)** *IFNB1* mRNA expression kinetics upon dsRNA electro-transfection was analyzed in A549 WT, A549 IRF3 KO, and A549 ROSA26-IRF3 (A549 IRF3 KO ROSA26-IRF3) expressing cells and used for model validation. Values were normalized to the housekeeping gene *GAPDH* and the 0 h time point subsequently using the

## IFN feedback upon RLR stimulation is essential for high and sustained ISG expression but not for IFN production

IFN exposure and signaling downstream of its cognate receptor through the JAK/STAT cascade does not induce the production of IFN in an auto-feedback manner (Fig S4A–E). However, it up-regulates numerous components required for PRR signaling, particularly RIG-I itself, thereby sensitizing cells for IFN production (indirect positive feedback). Here, we employed A549 cells with a double KO of type I (*IFNAR1*) and type III (*IFNLR*) IFNRs, termed IFNR double KO (A549 IFNR DKO), to further characterize RLR signaling dynamics in the presence or absence of IFN feedback (Fig S2C). We synchronously stimulated A549 WT and A549 IFNR DKO cells by electro-transfection with 5′ppp-dsRNA, and monitored type I and type III IFN induction (Fig 4A), and ISG mRNA expression (Fig 4B) over time. Interestingly, the dynamics of IFN mRNA induction upon stimulation were indistinguishable between both cell lines: after a rapid induction within the first hour of dsRNA stimulation and reaching peak levels between 6 and 8 h, IFN mRNA expression decreased again between 12 and 24 h poststimulation. These results further corroborated the notion that IFN production is dependent on RLR signaling and independent of IFN signaling (Fig 4A). Similarly, using a multiplex immunoassay, we analyzed IFN-*β* and IFN-*λ*1 secretion, which could be detected from 6 h post–electro-transfection onwards and plateaued at later time points with no notable difference between WT and IFNR DKO cells (Fig 4C). In contrast, we observed a substantial difference in the expression of the ISGs *IFIT1* and *MX1* (Fig 4B). Although initial induction up to 2 h (*MX1*) or 6 h (*IFIT1*) was unaffected, the lack of IFN feedback in IFNR DKO cells led to a significant decrease in *IFIT1* mRNA levels over the following hours and a substantially lower overall induction for *MX1* (Fig 4B). Also at the protein level, MX1 was not detected in dsRNA-stimulated A549 IFNR DKO cells, whereas WT cells showed increased expression from 12 h on (Fig 4D). In line with the IFN-dependent production of MX1, phosphorylated STAT2, a hallmark of active IFN signaling and ISGF3 formation, was undetectable in A549 IFNR DKO cells. Interestingly, key components of RLR signaling, such as RIG-I or CCL5 expression, IRF3 phosphorylation, and IkB*α* degradation, showed virtually identical kinetics in WT and IFNR DKO cells (Fig 4D). This corroborates that IFN production is solely dependent on RLR signaling and not (directly) dependent on feedback via IFNR signaling. Expression of the dsRNA sensor RIG-I itself was considerably up-regulated over time even in IFNR DKO cells and, thus, demonstrates that the RLR pathway exhibits an IFN-independent feedback regulation through which the pathway reinforces itself. In a context of virus infection, in which stimulatory dsRNA is not supplied as one transient pulse (as in transfection) but remains present or even increases over time because of replication, this positive auto-feedback of the pathway would lead to increasing signaling strength over time.

## The expanded RIG-I pathway model comprising IFN feedback reproduces the cell-intrinsic antiviral response

Having established a dynamic pathway model of the core RIG-I signaling module, we were able to accurately predict the pathway outcome from viral RNA recognition by RIG-I to transcription of IFN-*β* by IRF3 within the first few hours after RIG-I stimulation (Fig 3). However, so far, our model did not capture the translation and secretion of IFNs and, thus, could not recapitulate the effects of IFN signaling that considerably shape the antiviral state over the prolonged time of RLR stimulation (e.g., 24 h, Fig 4). To overcome this limitation, we extended our model by combining our core RIG-I signaling module with an ODE-based model of the type I IFN-triggered JAK/STAT-signaling pathway previously developed by Maiwald et al (2010). The latter is a detailed model of JAK/STAT signaling upon stimulation with IFN-*α*. To connect the two modules, we quantified secreted IFN levels upon RIG-I pathway stimulation and linked its production to the IFN-*β* mRNA levels. The produced IFN levels acted as input for the IFN-signaling module from the literature (Fig 5A and B). Our combined model comprises the cell-intrinsic antiviral response from incoming viral RNA to the expression of antiviral effector proteins downstream of IFN signaling. The JAK/STAT model introduced 66 additional species engaged in 41 reactions. Importantly, we fixed all rate constants to the values determined previously—for the RIG-I module as described above, and for the IFN-signaling module as determined by Maiwald et al (2010). The only rate constant that needed to be adjusted (fitted to the data) was $k_{68}$, linking the amount of produced IFN to the effective concentration triggering IFNAR signaling. Protein concentrations of components in the JAK/STAT pathway were updated to the quantities we determined by mass spectrometry for A549 cells. Without adjusting any further parameters, the model was able to accurately describe all measured pathway components over the experimental time frame of 24 h (Fig 5C). One component of the model that suffered high experimental variability was phosphorylation of STAT2 because of its high sensitivity to minimal concentrations of IFN, resulting in a very pronounced initial activation. To further account for a more detailed description of ISG expression, we extended the model to include the dynamics of selected ISG mRNAs and proteins. We introduced eight new species and updated the differential equation for RIG-I to account for its up-regulation (Supplemental Data 1).

The downstream production of antiviral effector proteins, such as MX1, could be reproducibly quantified experimentally and accurately captured by the model (Fig 5C). In their study, Maiwald and colleagues identified IRF9 as a central and rate-limiting component of IFN signaling (Maiwald et al, 2010). Hence, to further challenge and validate our model, we experimentally modulated IRF9 levels. Since in our A549 cell system IRF9 appeared not to be limiting (Fig S5), we decreased its expression by reconstituting it under a weak promoter (ROSA26-IRF9) in a CRISPR/Cas9-mediated KO (IRF9 KO) background (Fig S2E). We synchronously stimulated cells by electro-transfection of 5′ppp-dsRNA and measured the mRNA expression of *IFNB1*, *IFIT1*, and *MX1* over time (Fig 5D). Very similarly to IFNR DKO conditions (Fig 4), *IFNB1* mRNA levels were exclusively controlled by RLR downstream signaling and, hence, not sensitive to modulation of IRF9 levels, whereas *IFIT1* was slightly affected, and

$2^{-\Delta\Delta Ct}$ method. The graph depicts the mean ± SD of three biologically independent experiments. **(E)** Core model simulation of *IFNB1* mRNA expression in reduced IRF3 protein level conditions.

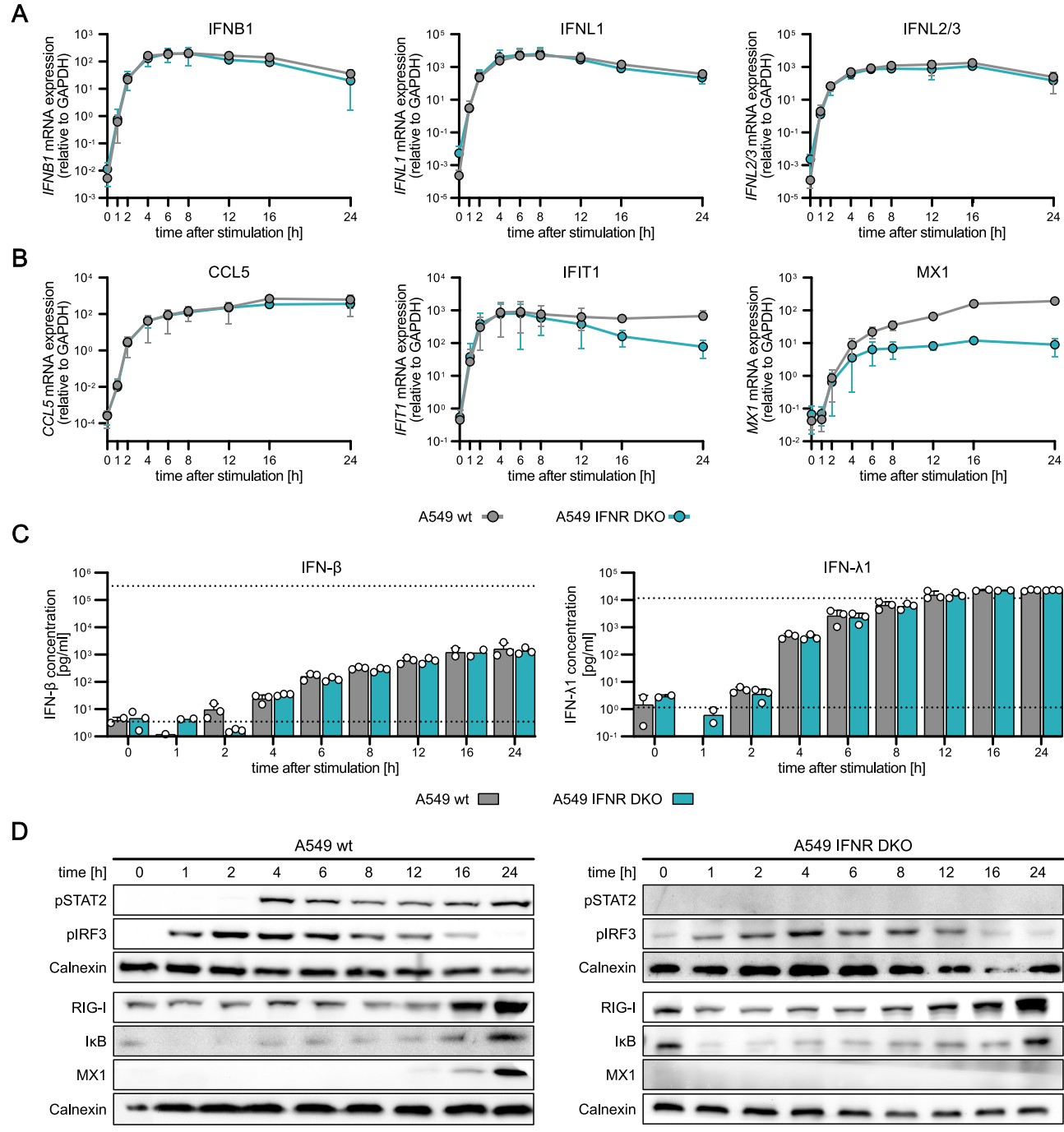

**Figure 4. RIG-I-signaling kinetics impact on IFN signaling.**
A549 WT and an A549 *IFNAR1 IFNLR* double KO cell line (IFNR DKO) were synchronously stimulated by electro-transfection with 220 ng in vitro generated 5′ppp-dsRNA. **(A, B)** *IFNB1*, *IFNL1* (IL-29), and *IFNL2/3* (IL-28) or (B) *CCL5*, *IFIT1*, and *MX1* mRNA expression kinetics were analyzed using quantitative RT–PCR. Values were normalized to the housekeeping gene *GAPDH* by applying the $2^{-\Delta Ct}$ method. **(C)** Secreted IFN-β and IFN-λ1 protein concentrations in pg/ml were determined using a multiplex immunoassay (U-PLEX IFN Combo, Meso Scale Diagnostics) in A549 WT and A549 IFNR DKO cells. Dashed lines indicate the upper and lower limits of quantification, respectively. **(D)** Protein abundance and phosphorylation status kinetics of key signaling components in A549 WT and A549 IFNR DKO cells using Western blot analysis. **(A, B, C, D)** Graphs depict the mean ± SD of three biologically independent experiments (A, B, C) or representative blots of two biologically independent experiments (D).

the "classical" ISG *MX1* was considerably impacted by lacking or reduced IRF9 (Fig 5D). In essence, these results indicate that IRF9 levels are critical for the expression of some genes (*IFIT1*, *MX1*), whereas others seem to be independent of IRF9 protein levels

(*IFNB1*). The measured outcome of this experimental perturbation of the signaling system was correctly predicted by our model, further corroborating its robustness and reliability (Fig 5E) (Maiwald et al, 2010)

**Figure 5. Dynamic pathway model of core RIG-I-signaling coupled to the IFN-signaling model.**

Extension of the RIG-I signaling pathway mathematical model to include IFN signaling by using quantitative mRNA expression, protein phosphorylation, and protein abundance data obtained upon synchronous stimulation with 220 ng in vitro generated 5'ppp-dsRNA. **(A)** Schematic depiction of the established core RIG-I signaling model coupled to a previously published mathematical model of IFN signaling (Maiwald et al, 2010). **(B)** Gene expression legend for different transcription factors used in the extended mathematical model. **(C)** Quantitative protein abundance, protein phosphorylation, and mRNA data were used to combine and calibrate the extended, dynamic model of the antiviral system. Dots represent quantitative data of two biologically independent experiments, and lines represent average model fit. **(D)** *IFNB1*, *IFIT1*, and *MX1* mRNA expression kinetics upon synchronous dsRNA stimulation was analyzed in A549 WT, A549 IRF9 KO, and A549 ROSA26 IRF9 (A549 IRF9 KO ROSA26-IRF9) expressing cells and used for model validation. Values were normalized to the housekeeping gene *GAPDH* and the 0 h time point subsequently using $2^{-\Delta\Delta Ct}$. Graphs depict the mean ± SD of three biologically independent experiments. **(E)** Coupled model simulation of *IFIT1* and *MX1* mRNA upon varying IRF9 protein levels.

## RIG-I signaling dynamics can be accurately modeled in different cell lines

For the model establishment, we focused on A549 cells for their well-known capacity for RLR and IFN signaling and their amenability to genetic modification. As innate immune signaling pathways are evolutionarily ancient and well-conserved, we tested our model's potential to be adapted to other cell types. For this purpose, we performed side-by-side 5'ppp-dsRNA stimulation of A549 cells and an unrelated liver (hepatocellular carcinoma) cell line, HepG2, and compared the ensuing mRNA expression (Fig 6A and B), protein abundance and phosphorylation (Fig 6C), and IFN secretion (Fig 6D) dynamics. Surprisingly, whereas *IFIT1* and *CCL5* mRNA expression resembled similar kinetics in both cell lines, IFN mRNA expression (Fig 6A) and IFN secretion (Fig 6D) were significantly reduced in HepG2. Accordingly, IFNR signaling-dependent STAT2 phosphorylation and subsequent RIG-I and MX1 mRNA and protein expression were strongly reduced in HepG2 (Fig 6B and C). Notably, although *IFIT1* mRNA expression kinetics (Fig 6A) was identical in both cell lines, protein expression was reduced in HepG2 cells compared with A549 (Fig 6C). We adapted our mathematical model to the new cell line by label-free mass spectrometry-based quantification of the main signaling components in HepG2 cells and adjusting the respective initial concentrations in our model (Supplemental Data 1). Strikingly, without any further adjustment of kinetic rate constants, the model was capable of adequately describing the mRNA kinetics of *IFNB1* and *MX1* in HepG2 cells (Fig 6E). Only experimental results of *IFIT1* mRNA kinetics did not fully resemble the reduced effect in model simulations (Fig 6E). This highlights the potential of our model to be adapted to different cell lines by only adjusting the baseline expression levels of signaling components.

Lastly, we wanted to verify the overall rapid pathway dynamics qualitatively in non-neoplastic primary cells and virus infection. For this purpose, we obtained human foreskin fibroblasts (HFF) and stimulated them by electro-transfection with 5'ppp-dsRNA or infection with vesicular stomatitis virus (VSV). Corroborating the highly similar kinetics of the RIG-I pathway across different cell types, we found rapid induction of *IFNB1*, *IFNL1*, and ISG mRNA and IFN-β secretion upon electro-transfection of dsRNA closely resembling the kinetics observed in A549 and HepG2 cells (Fig S6A–D). Notably, also VSV infection triggered rapid signaling and IFN induction (Fig S7A–E). This was true despite its endocytosis-dependent, hence, putatively slower uptake mechanism (rather resembling liposome-based transfection), most likely because we synchronized infection by using a high MOI (MOI 10) and a short inoculation time (30 min). These findings further substantiate the notion that RLR and IFN signaling are highly conserved and robust across different cell types. Moreover, the pathway dynamics we describe here are independent of the exact RIG-I ligand, be it in vitro purified 5'ppp-dsRNA or actual virus infection-associated RNAs.

## The impact of viral antagonists on antiviral signaling dynamics can be simulated by our mathematical model

Upon virus infection, the kinetics and magnitude of the ensuing IFN response critically determine the outcome of infection. Therefore, most viruses have evolved potent antagonists capable of inhibiting host–cellular antiviral responses (García-Sastre, 2017). To better understand these critical host–virus interactions, it is important to investigate viral immune antagonism at a dynamic level. Our modeling approach may offer a valuable tool for studying the impact of antagonists on the dynamics of ISG induction. To test this, we have chosen well-described viral proteins with different modes of action: NS3/4A of HCV proteolytically inactivates the central adapter MAVS (Li et al, 2005), N[pro] of CSFV triggers the degradation of IRF3 (Bauhofer et al, 2007; Seago et al, 2007), and NS5 of dengue virus (DENV) degrades STAT2 (Ashour et al, 2009; Mazzon et al, 2009). Whereas NS3/4A and N[pro] target RLR signaling and thereby IFN induction, NS5 blocks signaling downstream of the IFNR.

Using lentiviral transduction, we generated A549 cells stably expressing varying levels of the viral proteins. In the case of NS3/4A, already minute amounts sufficed for maximum MAVS cleavage. Therefore, we here used graded doses of simeprevir, a pharmacological compound specifically inhibiting NS3/4A protease activity (Rosenquist et al, 2014; Meewan et al, 2019), which allowed us to mimic the effect of decreasing NS3/4A activity. We stimulated A549 cells by electro-transfection of 5'ppp-dsRNA and measured *IFIT1* and *IFNB1* mRNA levels over 24 h (Figs 7 and S8A).

For NS3/4A, we observed a very efficient cleavage of MAVS (Fig 7B), leading to a substantial (30-fold) reduction of *IFIT1* mRNA levels, particularly at early time points (Fig 7A). Notably, a basal level of MAVS withstood NS3/4A expression (Fig 7B), leading to the remaining signaling and induction of *IFIT1* expression. As expected, increasing concentrations of simeprevir dose-dependently decreased NS3/4A activity leading to increased amounts of intact MAVS protein (Fig 7B). This dose-dependent effect was reflected in *IFIT1* and *IFNB1* kinetics, with increasing concentrations of simeprevir restoring induction to the level of the catalytically inactive NS3/4A S139A control (Figs 7A and S8A). Taken together, increasing NS3/4A activity reduced the expression of *IFIT1* and *IFNB1*. For *IFIT1*, this effect was less pronounced for later time points (12 and 24 h); nonetheless, the overall kinetics of induction was not altered significantly. Interestingly, these expression dynamics were also qualitatively predicted by our mathematical model when we systematically reduced MAVS concentrations in silico (Fig 7C), highlighting the utility of the modeling approach to investigate viral immune evasion.

N[pro] directs the transcription factor IRF3 for degradation (Fig 7E), and accordingly, we observed a near-complete inhibition of *IFIT1* and *IFNB1* induction for the highest N[pro] levels (Figs 7D and S8B). With decreasing expression of the viral antagonist, *IFIT1* and *IFNB1* induction dose-dependently recovered (Figs 7D and E and S8B). In contrast to the HCV protease, N[pro] considerably affected the kinetic profile of *IFIT1* and *IFNB1* induction, strongly repressing their early expression up to 8 h poststimulation (Figs 7D and S8B). We then used our model to simulate the N[pro] effect by gradually reducing IRF3 abundance in our model. The predicted kinetics of the antiviral response matched our experimental data at early time points (Fig 7F, compare with Fig 7C), and our results underscore the potency of N[pro] as an immune antagonist.

Next, we tested the effects of DENV NS5 expression, which targets STAT2 for degradation and therefore affects signaling downstream of the IFNR, but not IFN induction through the RLR pathway. Increasing levels of NS5 indeed led to decreasing amounts of STAT2,

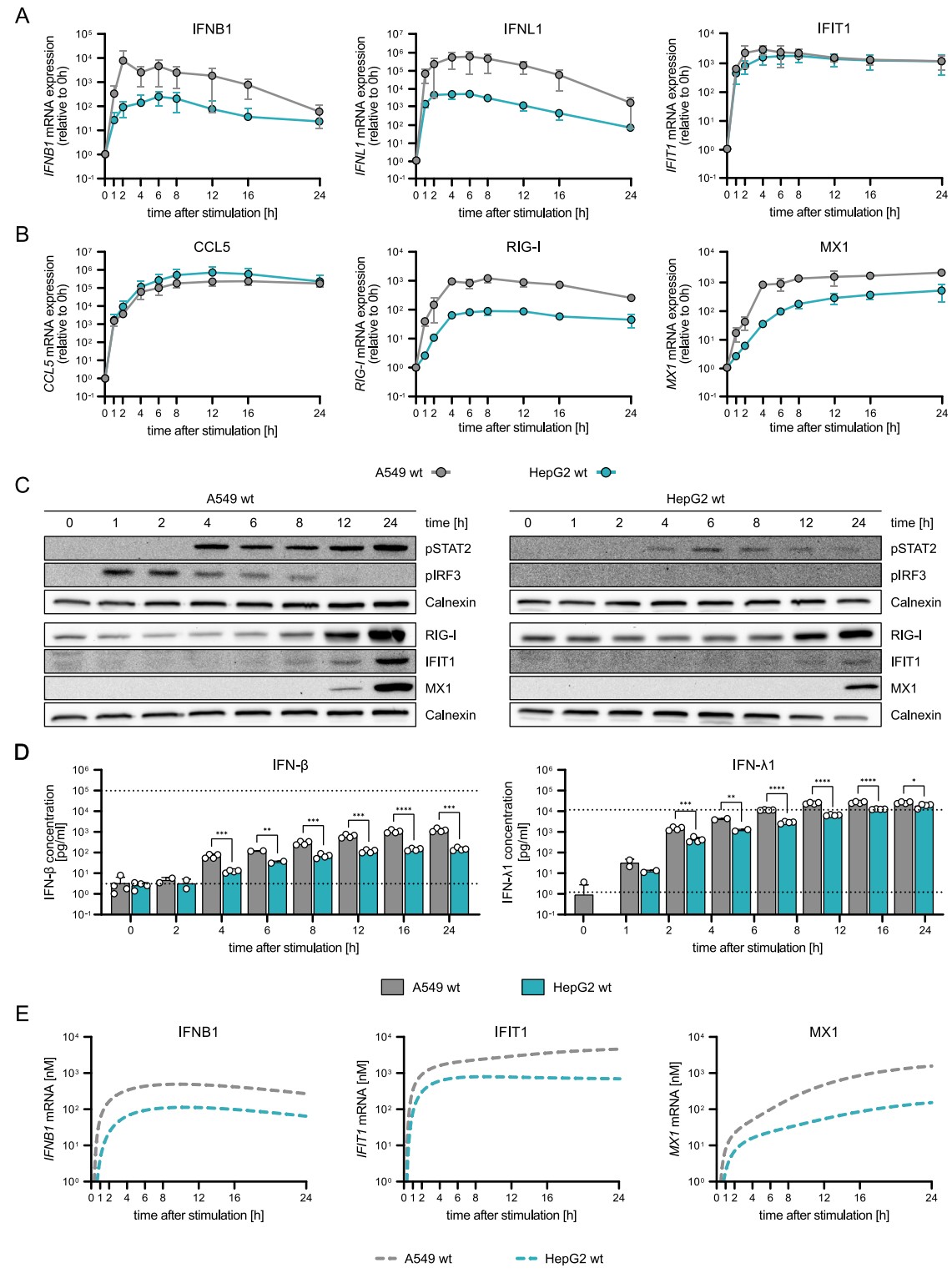

**Figure 6. Characterization of RIG-I-signaling dynamics upon synchronous dsRNA simulation in two unrelated cell lines.**
A549 WT and HepG2 WT cells were synchronously stimulated with 220 ng 5′ppp-dsRNA by electro-transfection and RNA, protein, and supernatants were harvested at different time points. **(A, B)** *IFNB1*, *IFNL1*, *IFIT1*, and (B) *CCL5*, *RIG-I*, and *MX1* mRNA expression kinetics in A549 WT and HepG2 WT cells was analyzed using quantitative RT–PCR. Values were normalized to the housekeeping gene *GAPDH* and the 0 h time point subsequently using $2^{-\Delta\Delta Ct}$. **(C)** Quantitative protein abundance and protein phosphorylation upon synchronous stimulation in A549 WT and HepG2 WT cells were analyzed using Western blot analysis. **(D)** Secreted IFN-$\beta$ and IFN-$\lambda$1 protein concentrations were determined using a multiplex immunoassay (U-PLEX IFN Combo, Meso Scale Diagnostics) in A549 WT and HepG2 WT cells. Dashed lines indicate the upper and lower limits of quantification, respectively. **(E)** Model predictions for *IFNB1*, *IFIT1*, and *MX1* mRNA expression upon synchronous stimulation of A549 WT and HepG2 WT cells. **(A, B, C, D)** Graphs depict the mean ± SD (A, B, D) or representative blots (C) of three biologically independent experiments. Statistical significance was determined with $t$ test (****, $P < 0.0001$; ***, $P < 0.001$; **, $P < 0.01$; *, $P < 0.05$).

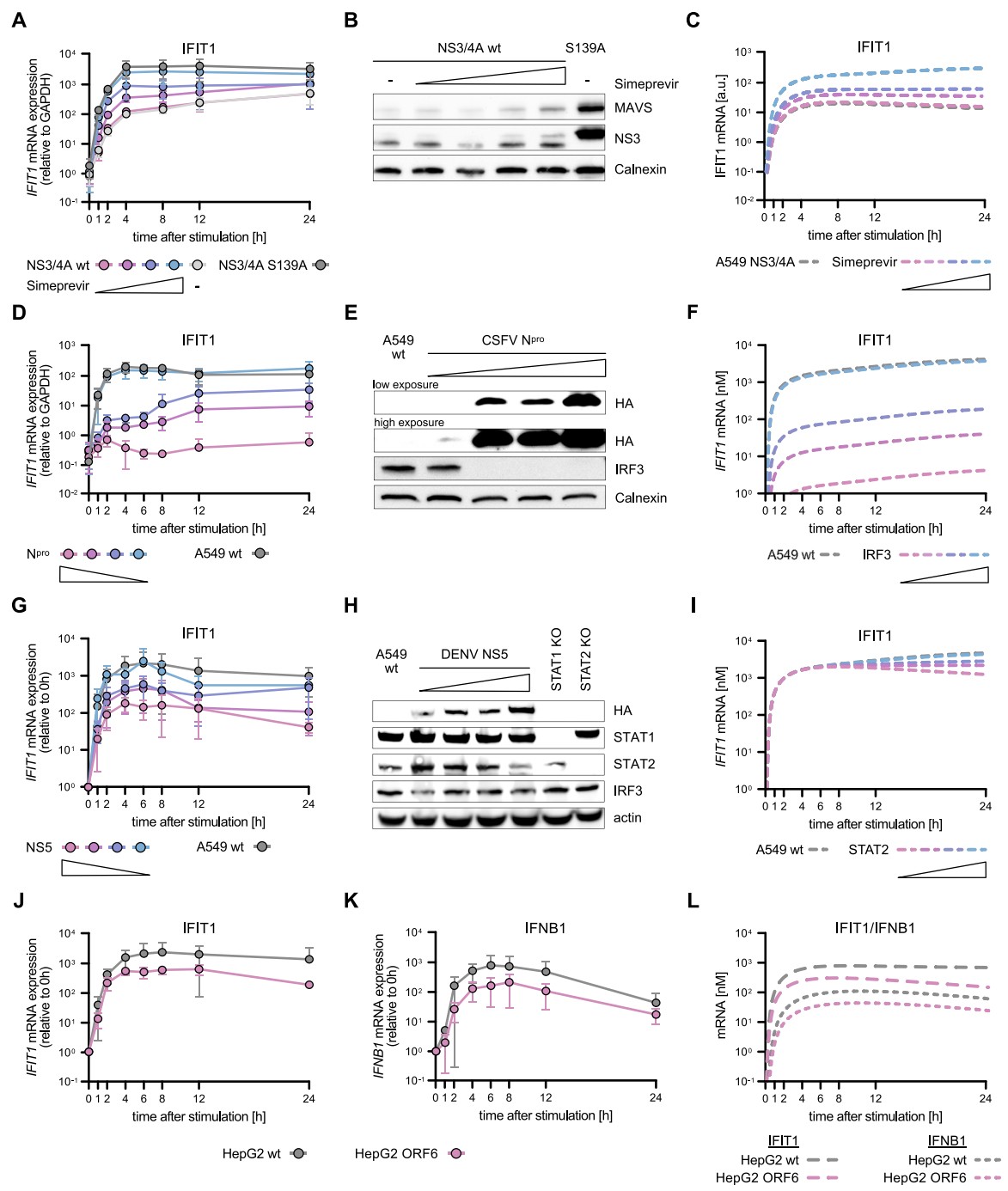

**Figure 7. Influence of various viral antagonists on RIG-I signaling dynamics in A549 and HepG2 cells.**
**(A, B, C, D, E, F, G, H, I, J, K, L)** A549 cells stably expressing the viral proteins NS3/4A of hepatitis C virus (HCV, (A, B, C)), N^pro of classical swine fever virus (CSFV, (D, E, F)), NS5 of dengue virus (DENV, (G, H, I)) or HepG2 cells stably expressing ORF6 of SARS coronavirus type 2 (SARS-CoV-2, (J, K, L)) were generated using lentiviral transduction. **(A, B, C)** A549 cells stably expressing either the viral HCV NS3/4A protein or a catalytically inactive version (NS3/4A S139A) were treated with increasing protease inhibitor (simeprevir) concentrations overnight and subsequently synchronously stimulated with 220 ng 5'ppp-dsRNA. **(A, B)** *IFIT1* mRNA expression kinetics and (B) basal NS3 and MAVS protein levels were analyzed in A549 NS3/4A WT and A549 NS3/4A S139A overexpression cell lines. **(C)** Model simulation of *IFIT1* mRNA kinetics upon synchronous stimulation with dsRNA in A549 cells expressing NS3/4A in the absence or presence of increasing amounts of simeprevir. **(D, E, F)** A549 WT cells and cells expressing different levels of HA-tagged CSFV N^pro were synchronously stimulated with 220 ng 5'ppp-dsRNA. **(D, E)** *IFIT1* mRNA expression kinetics and (E) basal HA-N^pro or IRF3 protein levels were analyzed in A549 WT or N^pro overexpression cell lines using qRT–PCR and Western blotting. **(F)** Model simulation of *IFIT1* mRNA kinetics upon synchronous stimulation with dsRNA with varying levels of IRF3. **(G, H, I)** A549 WT cells and cells expressing different levels of HA-tagged DENV NS5 were synchronously stimulated with 220 ng 5'ppp-dsRNA. **(G, H)** *IFIT1* mRNA expression kinetics and (H) basal HA-NS5, STAT1, STAT2, and IRF3 protein levels were analyzed in A549 WT, STAT1 KO, STAT2 KO or NS5 overexpression cell lines. **(I)** Model simulation of *IFIT1* mRNA kinetics upon synchronous stimulation with dsRNA with varying levels of STAT2. **(J, K)** *IFIT1* and (K) *IFNB1* mRNA expression kinetics were analyzed in HepG2 WT cells and cells expressing Strep-tagged SARS-CoV-2 ORF6 upon synchronous stimulation with 220 ng 5'ppp-dsRNA. **(L)** Model simulation of *IFIT1* and *IFNB1* mRNA expression upon synchronous stimulation. **(A, D, G, J, K)** qRT–PCR values were either only

but not STAT1 or IRF3 (Figs 7H and S2B). Of note, even for the highest concentration of NS5 STAT2, degradation was only partial (~40% remaining). Correspondingly, *IFIT1* expression was affected particularly at late time points when its transcription was predominantly driven by IFN signaling, leading to a decrease in mRNA levels from 8 h poststimulation onwards (Fig 7G). For the higher concentrations of NS5, early fold-induction of *IFIT1* was inhibited, which was unexpected and not predicted when we simulated *IFIT1* induction under conditions of limited STAT2 availability in our model (Fig 7I). In contrast, induction of *IFNB1* was unaffected by NS5, in line with a sole effect on STAT2 (Fig S8C). Hence, further studies will be needed to elucidate whether NS5 might have additional, so far unknown functions in the early stages of the antiviral response.

Finally, we used a viral antagonist with more than one defined target: ORF6 of the recently emerged SARS coronavirus 2 (SARS-CoV-2). ORF6 negatively impacts IFN induction and IFN signaling (Hayn et al, 2020 *Preprint*), mostly by blocking IRF3 (Xia et al, 2020; Yuen et al, 2020), STAT1, and STAT2 (Miorin et al, 2020) nuclear translocation (Minkoff & tenOever, 2023). In A549 cells, ORF6 expression did not impact *IFIT1* or *IFNB1* mRNA expression upon dsRNA stimulation but resulted in decreased luciferase expression in an *IFIT1*-promoter reporter assay (Figs S2D and S8D–G). By interfering with the nuclear import–export machinery, ORF6 has also been reported to inhibit the nuclear export of mRNA (Minkoff & tenOever, 2023), which might explain why luciferase protein production was affected, whereas gross mRNA levels were not (lysis does not discriminate nuclear and cytosolic mRNA). In contrast to A549 cells, in HepG2 cells, we observed a significant effect on both, transcript levels and luciferase production (Figs 7J and K, S2D, and S8H and I). Here, ORF6 affected *IFIT1* mRNA induction qualitatively similar to NS5, with a moderate reduction at early time points, but a clear decrease between eight and 24 h poststimulation. These findings are consistent with model simulations under conditions of simultaneously reduced IRF3 and STAT2 levels (Fig 7L).

In summary, we established an ODE-based mathematical model of antiviral signaling, comprising the RLR pathway of IFN induction and IFN signaling downstream of the IFNR. This model can very accurately simulate the activation of individual signaling components over time, and the induction of ISGs and the production of antiviral effector proteins. Importantly, it permits in silico simulation of viral interference with this cell-intrinsic defense system and offers a valuable tool to study the impact of yet unknown viral antagonists or other factors perturbing RLR and/or IFN signaling.

# Discussion

### Importance of cell-intrinsic immunity and the IFN system in infectious diseases

Evolutionary ancient innate immune responses (Kimbrell & Beutler, 2001; Buchmann, 2014) in humans represent the vital basis for all higher tiers of immune reactions. For one, cytokines produced and secreted at early stages of infection play a critical role in the proper launching of adaptive immune responses and their coordination (Jain & Pasare, 2017). But even beyond activating and coordinating the action of professional immune cells, immediate cell-intrinsic responses of the infected cells potently suppress microbial, particularly viral replication, which is frequently an absolute prerequisite for successful clearance of the pathogen (Dupuis et al, 2003). Recently, it has been shown for SARS-CoV-2 that improper functioning of the IFN system very strongly predisposes individuals to life-threatening COVID-19 (Zhang et al, 2020a, 2020b; Bastard et al, 2020), and we and others have shown that a particularly swift and strong IFN response in children associates with their resilience towards severe courses of the disease (Yoshida et al, 2021; Loske et al, 2022; Magalhães et al, 2023 *Preprint*). In general, the immune-driven pathogenesis of COVID-19 illustrates the critical importance of timely and well-coordinated cytokine production for a successful immune response, and the potential of a dysregulated IFN and cytokine system to cause substantial pathology (Wong & Perlman, 2021). Lastly, slowed-down or dampened IFN responses are observed in elderly individuals as part of the so-called immunosenescence, rendering aged patients more vulnerable to infectious diseases (Metcalf et al, 2015; Molony & Iwasaki, 2017), but also occur at conditions of lower ambient temperature, likely contributing to the seasonality of certain viral infections (Foxman et al, 2015; Prow et al, 2017; Lane et al, 2018). Hence, a proper understanding of the dynamics of cell-intrinsic immune response is essential.

### Rapid dynamics of IFN induction through the RIG-I/IRF3 pathway

The importance of pathway dynamics prompted us to optimize the experimental setup to characterize RIG-I signaling kinetics more precisely. A study by Thoresen et al (2022) recently reported on the very rapid and so far underestimated activation kinetics of the RIG-I pathway. Nonetheless, as in virtually all previous investigations, it was based on liposome-mediated transfection and virus infection, potentially introducing significant delays and stochasticity in cytosolic dsRNA uptake. In fact, we observed a striking difference in the activation kinetics of RIG-I signaling when we used electroporation-based millisecond transfection as opposed to classical liposome and endocytosis-based transfection. We observed a synchronous activation of all cells as determined by IRF3 nuclear translocation, occurring 15–30 min after stimulation. Whereas the artificial overexpression of IRF3 in this experiment might alter absolute translocation kinetics, timing was compatible with all other readouts we assessed (protein phosphorylation, mRNA transcription) and the striking qualitative difference of electro-transfected versus lipofected cells is unlikely to be affected by total IRF3 levels. The first transcripts of type I and III IFNs could be detected after 45–60 min, with measurable amounts of secreted IFNs in the culture supernatants after 4–6 h. Notably, also the study by Thoresen et al (2022) confirmed these rapid activation kinetics despite using liposome-based transfection and infection. Of note,

---

normalized to the housekeeping gene *GAPDH* (A, D) or, in addition, normalized to the 0 h time point (G, J, K). **(A, B, D, E, G, H, J, K)** Graphs depict the mean ± SD (A, D, G, J, K) or representative blots (B, E, H) of three biologically independent experiments.

the authors used high mass-wise concentrations of very short dsRNA, yielding a very high molarity for stimulation and thereby increasing the number of RNA molecules entering the cells at an early time. They demonstrated that signaling commences rapidly after minute amounts of stimulatory RNA reach the cytoplasm. Similarly, we find a very fast onset of signaling upon high titer, small inoculation volume VSV infection of HFF cells. In addition, a previous study by Bertolusso et al (2014)—also using 5′ppp-dsRNA electroporation of A549 cells and mathematical modeling (see below)—found comparably rapid signaling dynamics (IRF3 translocation, *IFNB1* transcription). However, in that study, the TLR3 pathway was reported to be co-stimulated and engaged in a novel antagonistic crosstalk with RIG-I signaling. Using CRISPR/Cas9-mediated functional KO of RIG-I or IRF3, we previously controlled for RIG-I-specificity and can rule out significant involvement of TLR3 or other receptors in our system (Willemsen et al, 2017; Wüst et al, 2021), and we have shown TLR3 expression and signaling is virtually absent in A549 cells (Plociennikowska et al, 2021).

**IFN production is independent of autocrine feedback through the IFN/JAK/STAT axis**

A central feature of the antiviral system is the positive feedback through autocrine IFN signaling. Albeit RLR signaling is known to directly induce certain antiviral effectors (ISGs), such as IFIT1 (Collins et al, 2004; Doğanay et al, 2017; Kell et al, 2020), the full-fledged transcriptional program mediating the antiviral state of a cell is only established downstream of IFNR signaling. Importantly, the RLRs and the IRF3-like transcription factor, IRF7, are ISGs substantially induced upon IFN signaling. Hence, in an actual viral infection, autocrine IFN signaling reinforces viral sensing and IFN production. In our experimental system, this increased RNA sensing and IFN production plays less of a role, as the dsRNA is only pulse-transfected and not replicated for prolonged periods of time as would be the case in an infection. Contrary to all previous studies, to the best of our knowledge, we, for the first time, experimentally uncoupled primary RLR signaling and auto/paracrine signaling through the IFNRs. When we used "IFN-blind" cells, harboring a functional double KO of the type I and type III IFNRs (IFNR DKO), the induction dynamics of IFNs were not affected at any time point tested. This corroborates the recent study by Thoresen et al (2022) in which pretreating cells with cycloheximide blocking any new protein synthesis did not impact the dynamics of IFN induction upon RIG-I stimulation. In contrast, IFIT1, being induced by both IRF3 and IFN signaling (Collins et al, 2004; Doğanay et al, 2017; Kell et al, 2020), was affected only at later time points, and MX1, canonically strictly dependent on IFN signaling (Ronni et al, 1998; Haller & Kochs, 2002; Schoggins et al, 2011; Haller et al, 2015), was substantially impacted throughout. Nevertheless, MX1 was still significantly induced even in the absence of IFN signaling, which has not been broadly appreciated previously but has been shown in human fibroblasts upon HCMV infection (Ashley et al, 2019). The strict exclusion of auto and paracrine signaling also sets our study apart from Bertolusso et al (2014), in which RIG-I or IRF3 knockdown by siRNAs did not lead to a significant reduction in ISG (e.g., RIG-I) induction upon dsRNA stimulation. As an alternative to the authors' hypothesis of TLR3 and IRF7 being co-stimulated, we propose this may have been an effect of incomplete knockdown, amplified by paracrine IFN signaling.

**A mechanistically interpretable model of RIG-I-initiated antiviral signaling**

We used literature knowledge of the core RIG-I pathway topology to set up a mathematical model based on a set of ODEs mechanistically representing all main canonical signaling steps. We have previously modeled dsRNA recognition by RIG-I and its downstream signaling, however, with a focus on the quantitative output rather than signaling kinetics (Schweinoch et al, 2020). Also, Bertolusso et al (2014) modeled RIG-I signaling using experimental measurements to fit their model. However, as discussed above, that study assumed co-stimulation of TLR3 and an intricate interaction of TLR3 and RIG-I signaling, which we could not confirm in our system. Therefore, we focused on modeling the RIG-I/IRF3 signaling axis only. Notably, Bertolusso et al (2014) found significant decreases in the protein levels of RIG-I upon dsRNA stimulation and speculated that this may be signaling-induced proteolytic degradation as a means of negative self-regulation of the pathway. In fact, we also observed slight decreases in RIG-I protein levels, particularly in IFNR DKO cells. Furthermore, we previously described inhibitory phosphorylation of RIG-I by DAPK1 as one amongst several reported negative feedback mechanisms (Willemsen et al, 2017; Onomoto et al, 2021). Although negative feedback is likely crucial in regulating cell-intrinsic antiviral signaling on longer time scales, our mathematical model described all signaling dynamics satisfactorily even in the absence of such negative feedback loops. Very likely, this is because of the transient pulse of stimulatory dsRNA in our system, leading to ceasing RIG-I activity because of a lack of ligand before negative regulation of the receptor occurs.

In contrast to negative feedback regulation of RIG-I, positive feedback (and feedforward) through IFN must be taken into account, in particular, for timeframes of more than 4 h (after which secreted IFN can be detected). Therefore, we coupled our mathematical model to a previously published model of type I IFN signaling (Maiwald et al, 2010). We took the IFN produced as the main output of our core RIG-I module and used it as input for the IFN-signaling module, only applying one fitted scaling factor. The IFN model is also based on ODEs and was trained on quantitative immunoblotting data, similar to our RIG-I model. However, experimentally, that study used IFN-α for stimulation, which is not produced downstream of RLRs in our cell system (Wüst et al, 2021). Nonetheless, it is a type I IFN much like IFN-β and potentially does not affect signaling dynamics to a relevant extent. A further difference to our system is their employed cell line, the human hepatoma cell line Huh7.5. Reassuringly, all kinetic rate constants of the published IFN model could still be employed in our combined model of the antiviral pathway. Upon only adjusting the initial protein concentrations to those measured in our A549 cells, the combined model of RIG-I and IFN signaling very accurately described our measured 24 h time course data. This impressively highlights the substantial degree of conservation of this pathway across different cell types (Kok et al, 2020).

## High conservation of signaling characteristics across different cell lines and types

To further corroborate the cell line independence of the overall rapid signaling dynamics and of our pathway model, we validated it using time-course measurements in the unrelated HepG2 cell line. Again, upon only adjusting the cell line-specific initial protein concentrations, our combined model was able to adequately predict RLR- and IFN-signaling dynamics in HepG2 cells upon electro-transfection of 5'ppp-dsRNA. We further confirmed the overall response dynamics qualitatively in human primary fibroblasts (HFF) upon dsRNA electro-transfection and virus infection, representing a completely unrelated cellular system more closely resembling physiological conditions. In light of the observed high degree of qualitative conservation of pathway dynamics, we expect that our model is generalizable to a wide variety of different cell lines and types by simply adjusting the basal protein concentrations of the proteins involved in signaling. This advertises the model as a valuable tool to study kinetic aspects of antiviral signaling even with experimental measurements only at a limited number of time points. We cannot rule out, however, that in some cell types (e.g., myeloid cells of the professional immune system) further regulatory systems not represented in the model might be in place, or additional transcription factors of (partially) overlapping function may exist that require modifications to the model topology. One such example is IRF7, a transcription factor highly similar to IRF3, which is expressed to high levels in plasmacytoid dendritic cells, rendering them competent to produce excessive amounts of IFN, including IFN-α, even before the positive feedback through the JAK/STAT pathway (Marié et al, 1998; Sato et al, 1998, 2000).

## Relevance of negative and positive feedbacks and feedforward regulation

As discussed above, negative feedback of the pathway was neglected in the present model as it was not required to accurately describe signaling dynamics in the chosen experimental system. Nonetheless, it is intriguing to contemplate its consequences and relevance. Within virus-infected cells, positive autocrine feedback via IFN signaling increases RIG-I levels, whereas negative feedback via triggered proteolysis or posttranslational modifications counteracts this (Onomoto et al, 2021); the net effect remains to be determined in future studies. On the contrary, positive feedforward via paracrine IFN signaling in noninfected cells (i.e., bystander cells with inactive RIG-I signaling) can substantially increase those cells' responsiveness by increasing RIG-I levels (RLRs in general) and other rate-limiting factors such as IRF9 (Maiwald et al, 2010; Kok et al, 2020). In fact, we have recently shown that such feedforward ("priming") can decisively alter cells' responses to rapidly replicating viruses with strong immune antagonists, such as SARS-CoV-2, by significantly reducing the latency in triggering the antiviral response (Kok et al, 2020; Loske et al, 2022; Magalhães et al, 2023 Preprint).

## Impact of viral antagonists on the dynamics of cell-intrinsic antiviral signaling

Pathogenic viruses have evolved elaborate strategies to evade the host immune response, including the cell-intrinsic IFN response

(García-Sastre, 2017). Most studies focus on the degree of IFN inhibition at a fixed (late) time point in the context of overexpression of a viral antagonist. In a natural infection, viral proteins accumulate over time and may lead to a delay rather than mere dampening of the IFN response. We propose that our dynamic pathway model offers a powerful framework to study such dynamic virus–host interactions. As a proof of principle, we have selected four well-known viral proteins interfering at defined steps with the host–cell antiviral defense: the proteases of HCV (NS3/4A) and CSFV (N$^{pro}$) both targeting RLR signaling and, thus, inhibiting the induction of IFN, dengue virus NS5 interfering with IFN signaling, and the ORF6 protein of SARS-CoV-2, exhibiting a multi-pronged strategy to inhibit the antiviral system. NS3/4A very efficiently cleaves and inactivates MAVS (Li et al, 2005). We found its activity to have an overall dampening effect on IFIT1 induction, most pronounced for the early time points, coherent with its targeting of the early induction phase of the antiviral response. CSFV N$^{pro}$ directs IRF3 for proteasomal degradation (Bauhofer et al, 2007; Seago et al, 2007), leaving NF-κB signaling unhindered and, thus, allowing for the production of certain pro-inflammatory cytokines (Summerfield & Ruggli, 2015; Fan et al, 2018; Khatoon et al, 2019). We found a substantially stronger impact of N$^{pro}$ on early IFIT1 induction than with NS3/4A. When we simulated the effects of these two viral antagonists in the model, it very closely captured their impact, underscoring the utility of the model to predict the effects of viral antagonists. Dengue virus NS5 targets signaling downstream of the IFNRs by leading to the degradation of STAT2 (Ashour et al, 2009; Mazzon et al, 2009). Accordingly, we observed no impact on IFNB1 expression, and a clear decrease of IFIT1 levels between 8 and 24 h. This matches the timing when IFIT1 transcription is largely shifted from the early IRF3 to the IFN-dependent transcription factor ISGF3, comparable with the effects we observed in IFNR DKO or IRF9 KO conditions. Nonetheless, NS5 expression had an effect on IFIT1 induction at time points before 8 h, too, which could not be explained by simulating a pure STAT2-mediated effect in our model. This might be a consequence of diminished STAT2 levels impacting the basal expression of members of the RLR-signaling pathway; however, this is arguable given by the unaltered IFNB1 dynamics. Alternatively, it may suggest DENV NS5 has additional targets in the induction phase of the antiviral defense at or downstream of the RLRs. One previous study proposed an interaction of NS5 and the Daxx protein (Khunchai et al, 2012), competing with the NF-κB/Daxx interaction and, hence, leading to NF-κB release and expression of CCL5 (Khunchai et al, 2012; Sahili & Lescar, 2017). It will be interesting to follow up on this in future studies. Lastly, we expressed SARS-CoV-2 ORF6, a viral antagonist inhibiting IFN induction and IFN signaling (Hayn et al, 2020 Preprint; Minkoff & tenOever, 2023). Upon ORF6 expression in HepG2 cells, we observed a very modest impact at time points before 4 h but significantly reduced peak levels of IFIT1 mRNA and decreased transcript levels between 8 and 24 h. This confirms that ORF6 has several targets in the antiviral system (Hayn et al, 2020 Preprint), but indicates the dominant effect may be at or downstream of IFN production. This was corroborated by modeling, where inhibition of ISGF3 formation approximated the experimental data much closer than limiting IRF3 activity (Fig 7L, compare with Fig 7F). In A549 cells, we found no impact of ORF6 on IFIT1 or IFNB1 transcript levels, nonetheless, translation of a reporter protein

(luciferase) under the control of the *IFIT1* or *IFNB1* core promoter was decreased significantly, comparable with HepG2 cells. Hence, we propose ORF6's potential to interfere with the nuclear translocation of transcription factors, such as IRF3 (Xia et al, 2020; Yuen et al, 2020), STAT1, and STAT2 (Miorin et al, 2020), to be less impactful than its capability to block nuclear export of mature (antiviral) mRNAs (Minkoff & tenOever, 2023).

As a future perspective, it will be highly exciting to combine our comprehensive model of innate immune signaling with models of viral replication to gain a better mechanistic understanding of these intricate virus–host interactions decisive for the development of disease: viral RNA, amplified rapidly during viral replication, is the inducer of antiviral signaling; antiviral signaling, via IFN, will potently suppress viral replication (i.e., suppress the production of stimulatory RNA); viral RNA at the same time is translated to proteins inhibiting antiviral signaling. We have previously approached modeling the interaction between HCV and innate immune signaling, although at a much lower level of mechanistic detail (Zitzmann et al, 2020). Others have modeled pathogen–host interaction at various levels as well (Bocharov et al, 2010; Qiao et al, 2010; Cai & Yu, 2020; Talemi et al, 2021; Wang et al, 2021). Employing our molecularly detailed model of antiviral signaling will enable a substantially higher degree of interpretability and applicability of such approaches in the future.

### Summary

We present a highly accurate characterization of the dynamics of cell-intrinsic innate immune signaling to virus infection. Using IFN-blind cells, we were able to dissect the induction phase downstream of RLRs and the effector phase downstream of IFNRs, which are normally tightly linked and overlapping. We further used our quantitative time-series data to set up and calibrate the—to the best of our knowledge—most comprehensive mechanistically interpretable dynamic pathway model of the RLR/IFN-signaling network. This model very accurately predicts the kinetics of signaling events downstream of RNA recognition by RIG-I, including the feedback and broadening of the response by secreted IFN and JAK/STAT signaling. Owing to its mechanistic detail, the model is capable of simulating viral immune antagonism and, vice versa, can be used to investigate the mechanism of action of novel virus-encoded inhibitors. Combining models such as ours with models of virus infection and replication will open up new and powerful avenues for studying the dynamics of virus–host interactions.

## Materials and Methods

### Cell culture and cell line generation

A549, HepG2, and HEK 293T cell lines and primary HFFs were cultured in DMEM (high glucose; Life Technologies), supplemented with 10% (vol/vol) or 15% (HFFs) (vol/vol) FCS (Thermo Fisher Scientific), 1x nonessential amino acids (Thermo Fisher Scientific), 100 U/ml penicillin, and 100 ng/ml streptomycin (Life Technologies)

at 37°C, 95% humidity, and 5% $CO_2$. Lentiviral transduction was used for the generation of A549 and HepG2 overexpression (OE) cell lines and for A549 KO cell line generation. The CRISPR/Cas9 technology was used for stable KO generation in A549. DNA oligonucleotides coding for guide RNAs (listed in Table S1) against the respective genes were designed with e-crisp.org (Heigwer et al, 2014) and cloned into the expression vector LentiCRISPRv2 (Feng Zhang, #52961; Addgene). Lentiviral particles were produced by transiently transfecting HEK 293T cells with a total of 15 µg DNA consisting of the plasmids pCMV-dr8.91 (coding for HIV gag-pol), pMD2.G (coding for the VSV-G glycoprotein), and a lentiviral pWPI vector (coding for the protein of interest) or the expression vector LentiCRISPRv2 (coding for guide RNA) in a 3:1:3 ratio for 8 h using calcium phosphate transfection (CalPhos Mammalian Transfection Kit; Takara Bio Europe). 2 d after transfection, the supernatant was harvested, sterile-filtered, and used for the transduction of target cells. Transduction was carried out two times for 24 h and transduced cells were selected with appropriate antibiotics (5 µg/ml blasticidin; MP Biomedicals; 1 µg/ml puromycin; Sigma-Aldrich; or 1 mg/ml geneticin (G418); Santa Cruz). Single-cell clones of successfully transduced cells were isolated, and KO or OE was validated by immunoblotting and if appropriate, functional tests. A549 IRF3 KO, IRF9 KO, STAT1 KO, and IRF3-eGFP H2B-mCherry were reported previously (Urban et al, 2020; Wüst et al, 2021; Zander et al, 2022).

### Live-cell imaging and quantification of IRF3 nuclear translocation

A549 cells stably expressing histone H2B-mCherry and eGFP-tagged IRF3 were seeded at a density of six x $10^4$ cells per 24-well. The next day, cells were stimulated with in vitro transcribed and chromatographically purified 400 bp 5'ppp-dsRNA (Binder et al, 2011) or poly(C) (Sigma-Aldrich) using Lipofectamine 2000 (Invitrogen) following the manufacturer's protocol (lipoplexes were left on cells through the duration of the experiment) or an in-well electro-transfection approach (Lonza). For the electro-transfection–based transfection of adherent cells, the 4D-Nucleofector Y Unit (Lonza), the AD2 4D-Nucleofector Y Kit (Lonza), and a homemade cytomix (120 mM KCl, 0.15 mM $CaCl_2$, 10 mM $KPO_4$, 25 mM HEPES, 2 mM EGTA, 5 mM $MgCl_2$, pH 7.6, added directly prior usage: 2 mM ATP, 5 mM GT) were used. First, the provided dipping electrode array was used for a mock transfection to decrease the electro-transfection intensity to maximize cell survival and transfection efficiency. Next, DMEM was replaced with 350 µl cytomix and the electrode was inserted into the 24-well plate strictly avoiding the formation of air bubbles. Electro-transfection was performed using the FB-166 program and cytomix was replaced with warm DMEM subsequently. IRF3-GFP nuclear translocation and H2B-mCherry colocalization were analyzed using confocal microscopy equipped with an incubation chamber (Olympus FluoView FV1000) or monitored in short time periods using a 10x magnification in an IncucyteS3 Live-Cell Analysis System (Sartorius AG). Image analysis was conducted with the machine learning software ilastik (Berg et al, 2019) using pixel and object classification to distinguish background from cell nuclei and to determine IRF3-GFP and H2B-mCherry colocalization, respectively. Calculations for object classification were performed by applying a size range of 60–500 and a threshold 0.85. Here, at

least 500 and up to 2,500 individual cells were analyzed for each time point in each condition.

## Synchronous stimulation using electro-transfection

Synchronous stimulation of the RIG-I pathway in A549- or HepG2-based cell lines was carried out using the Gene Pulser Xcell modular electro-transfection system (Bio-Rad) and a ShockPod cuvette chamber (Bio-Rad). Cell suspensions containing $4 \times 10^6$ cells were centrifuged at 700$g$ for 5 min, resuspended in 400 $\mu$l homemade cytomix, and transferred to a 0.4 cm cuvette already containing 220 ng 5′ppp-dsRNA. Electro-transfection was performed at 150 V with an exponential decaying pulse for 10 ms. Transfected cell suspensions were directly transferred to prewarmed, antibiotic-free DMEM, washed twice in DMEM, and finally resuspended in 9.6 ml DMEM. Cells were seeded on six-well plates using 1.2 ml washed cell suspension per well and time point.

## Infection of HFFs

A total of $5 \times 10^5$ HFFs were seeded in the wells of a six-well plate. The following day, the cells were infected on ice by replacing the cell culture supernatant with fresh medium containing (VSV GFP*MQ, gently provided by Gert Zimmer; University of Bern), resulting in a MOI of 10. To enhance and synchronize the infection, the plates containing cells were centrifuged at 805$g$ and 4°C for 30 min before transferring to 37°C with 5% $CO_2$. After 1 h, cells were washed with citric acid pH 5 for 5 min to remove any unbound viruses. After a subsequent wash with PBS, the cells were cultured with fresh medium, marking the start of the experiment (time point 0).

## Quantitative immunoblotting

Synchronously stimulated cells were in-well lysed with 100 $\mu$l Laemmli buffer (16.7 mM TRIS pH 6.8, 5% glycerol, 0.5% SDS, 1.25% $\beta$-mercaptoethanol, 0.01% bromophenol blue), containing Benzonase Nuclease for 10 min at room temperature, and subsequently incubated at 95°C for 5 min. Protein extracts were separated on 8–12% polyacrylamide gels by SDS–PAGE and transferred to PVDF membranes (0.2 $\mu$m pore size; Bio-Rad) in a wet-transfer approach using the Mini Trans-Blot cell (Bio-Rad). Membranes were blocked in PBS-T or TBS-T complemented with 5% (wt/vol) BSA (A3294-50G; Sigma-Aldrich) for up to 3 h at room temperature. Subsequently, the membranes were incubated with PBS-T or TBS-T complemented with 5% BSA and primary antibodies specific for $\beta$-actin (A5441, 1:2,000; Sigma-Aldrich), calnexin (ADI-SPA-865-F, 1:1,000; Enzo Life science), HA (H3663, 1:1,000; Sigma-Aldrich), IFIT1 (H00003434-DO1, 1:1,000; Abnova), I$\kappa$B$\alpha$ (9242, 1:1,000; Cell Signaling), IRF3 (sc-9082, 1:1,000; Santa Cruz), IRF9 (ab126940, 1:1,000; Abcam), MAVS (ALX-210-929-C100, 1:1,000; Enzo Life science), MX1 (kind gift of Georg Kochs, 1:1,000), NS3 (NS3B, kind gift of Darius Moradpour, 1:1,000), phospho-IKK$\alpha$/$\beta$ (2078, 1:1,000; Cell Signaling), phospho-IKK$\varepsilon$ (06-1,340, 1:1,000; Millipore), phospho-IRF3 (4947, 1:1,000; Cell Signaling), phospho-NF-$\kappa$B (3033, 1:1,000; Cell Signaling), phospho-STAT2 (D3P2P, 1:1,000; Cell Signaling), phospho-TBK1 (ab109272, 1:1,000; Abcam), RIG-I (AG-20B-0009, 1:1,000; Adipogen), STAT1 (610115, 1:

1,000; BD) or STAT2 (sc-514193, 1:1,000; Santa Cruz) at 4°C overnight. Furthermore, membranes were incubated with anti-rabbit HRP (A6154-5X1ML, 1:20,000; Sigma-Aldrich) or anti-mouse HRP (A4416-5X1ML, 1:10,000; Sigma-Aldrich) for 1 h at room temperature. For detection, Amersham ECL Prime Western Blotting Detection Reagent (Thermo Fisher Scientific) was applied for 1 min and luminescence was detected using the INTAS ECL ChemoCam Imager 3.2 (INTAS Science Imaging Instruments). Western blot bands were analyzed and quantified using Image J (1.52e).

## RNA isolation, reverse transcription, and quantitative PCR

RNA isolation (New England Biolabs), reverse transcription (High-Capacity cDNA Reverse Transcription Kit; Thermo Fisher Scientific), and quantitative real-time PCR (qRT–PCR; iTaq Universal SYBR Green Supermix; Bio-Rad) were performed according to the manufacturer's protocols, and qRT–PCR was performed on a CFX96 real-time system (Bio-Rad). Sequences of specific exon-spanning PCR primers are listed in Table S2. Values were normalized to the housekeeping gene GAPDH using the $2^{-\Delta Ct}$ method or fold changes were calculated relative to the 0-h time point using the $2^{-\Delta\Delta Ct}$ method (Schmittgen & Livak, 2008).

## Multiplex immunoassay

Quantitative measurement of secreted cytokines within supernatants was performed using the multiplex Meso Scale Diagnostics (MSD) platform. Briefly, supernatants of synchronously stimulated A549 WT, A549 IFNR DKO, and HepG2 WT cells were harvested, centrifuged, and multiplex assays were conducted according to the manufacturer's instructions. The analytes IFN-$\beta$ and IFN-$\lambda$1 were measured using the human U-PLEX IFN Combo (15094K-1; Meso Scale Diagnostics LLC). Plate readout was conducted using the MESO QuickPlex SQ 120 instrument (Meso Scale Diagnostics) and measurements were evaluated using the MSD Discovery Workbench software.

## ELISA

IFN-$\beta$ production of VSV-infected HFFs was detected using the LumiKine human IFN-$\beta$ ELISA kit (#lumi-hifnbv2; Invivogen) according to the manufacturer's instructions. Lucia luciferase signals were assessed in a Mithras plate reader (Berthold Technologies GmbH & Co. KG).

## Luciferase reporter assay

$1.5 \times 10^5$ cells per well of a 24-well plate were seeded. The next day, the medium was aspirated and replaced with penicillin/streptomycin-free DMEM. 75 ng Firefly luciferase reporter and 25 ng Renilla luciferase reporter were transfected using Lipofectamine 2000 (Invitrogen) following the manufacturer's protocol. After 8 h, the medium was exchanged and cells were stimulated with SeV (MOI 0.004) for 16 h. Cells were washed with 500 $\mu$l PBS, lysed with 100 $\mu$l luciferase lysis buffer (10% glycerol, 1% Triton X-100, 25 mM glycylglycin, 15 mM $MgSO_4$, 4 mM EGTA, 1 mM DTT), and stored for at least 15 min at −80°C. Lysates were thawed and Firefly

and Renilla luciferase signals were directly measured in a Mithras plate reader using 400 $\mu$l Firefly assay buffer (25 mM glycylglycin, 15 mM KPO$_4$ buffer, 15 mM MgSO$_4$, 4 mM EGTA, 2 mM ATP, 1 mM DTT, 80 $\mu$M D-Luciferin) and Renilla luciferase assay buffer (25 mM glycylglycin, 15 mM KPO$_4$ buffer, 15 mM MgSO$_4$, 4 mM EGTA, 2 mM ATP, 1 mM DTT, 3.36 $\mu$M Coelenterazine) per well. For multiple well measurements, 100 $\mu$l 10% SDS was used to quench the light re-action in between well measurements. Renilla luciferase is con-stitutively expressed and thus serves as a control. Samples were measured in technical replicates and Firefly luciferase values were divided by Renilla luciferase values.

### Protein copy number estimation of A549 and HepG2 cells by total proteome analysis using LC–MS/MS

A549 cells in quintuplicates and HepG2 cells in quadruplicates were lysed in SDS lysis buffer (4% SDS, 10 mM DTT in 50 mM Tris–HCl pH 7.5, cOmplete protease inhibitor cocktail), boiled at 95°C for 8 min and sonicated for 15 min at 4°C (Bioruptor). Alkylation of proteins was performed for 20 min in the dark by the addition of 55 mM iodoacetamide in 50 mM Tris/HCl (pH 7.5), followed by normali-zation of the protein concentration (Pierce 660 nm Protein Assay). For each replicate, 50 $\mu$g proteins were precipitated with 4 vol/vol of prechilled acetone at –20°C overnight, pelleted by centrifugation (15,000$g$, 10 min, 4°C), washed with 80% (vol/vol) acetone, air-dried, and resuspended in 40 $\mu$l thiourea buffer (6 M urea, 2 M thiourea in 10 mM HEPES, pH 8.0). Proteins were pre-digested by the addition of 1 $\mu$g LysC in 40 $\mu$l 50 mM ammonium bicarbonate buffer (pH 8.0) for 3 h at 25°C. Digestion was completed with 1 $\mu$g Trypsin in 120 $\mu$l 50 mM ammonium bicarbonate buffer (pH 8.0) for 16 h at 25°C. After purification on C18 StageTips (HepG2 cells) or fractionation of peptides on SCX (seven fractions: flow-through, pH 11, pH 8, pH 6, pH 5, pH 4, pH 3) and purification on C18 StageTips (A549 cells) as described previously (Rappsilber et al, 2007), peptides were loaded onto a 50-cm reverse-phase analytical column (75 $\mu$m column diameter; ReproSil-Pur C18-AQ 1.9 $\mu$m resin; Dr. Maisch) and sep-arated using an EASY-nLC 1,200 system (Thermo Fisher Scientific). A binary buffer system consisting of buffer A (0.1% formic acid in H$_2$O) and buffer B (80% acetonitrile, 0.1% formic acid in H$_2$O) with a 120-min (A549 cells) gradient (5–30% buffer B for 95 min, 30–95% buffer B for 10 min, wash out at 95% buffer B for 5 min, decreased to 5% buffer B for 5 min, and 5% buffer B for 5 min) was used at a flow rate of 300 nl/min. In contrast, peptides derived from HepG2 cells were separated with a 180-min gradient. Eluting peptides were directly analyzed on a Q-Exactive HF mass spectrometer (Thermo Fisher Scientific) operated in data-dependent acquisition mode with re-peating cycles of one MS1 full scan (300–1,650 m/z, R = 60,000 at 200 m/z) at an ion target of 3 × 10$^6$, followed by 15 MS2 scans of the highest abundant isolated and higher-energy collisional disso-ciation fragmented peptide precursors (R = 15,000 at 200 m/z). Peptide precursor isolation for MS2 scans was, in addition, lim-ited by a maximum injection time of 25 ms and an ion target of 1 × 10$^5$, whereas repeated isolation of the same precursor was dy-namically excluded for 20 s. Spectra were processed with MaxQuant (A549: version 1.6.6.0; HepG2: 1.6.10.43) using label-free quantifi-cation, match between runs, fixed carbamidometylation, and var-iable N-acetylation and methionine oxidation. Peptides and

proteins were identified by searching against the human proteome sequences (3.2016; UniprotKB) controlled by a false discovery rate of 0.01 (Tyanova et al, 2016a). Further analyses were performed with Perseus (version 1.6.6.0) (Tyanova et al, 2016b). Proteins only identified by site, matching the reverse sequence, or annotated as contaminants were excluded from the analysis. Protein copy numbers were estimated using the Proteomic Ruler plugin for Perseus as previously described (Wiśniewski et al, 2014). Briefly, raw protein intensities were separately averaged for each column and the histone proteomic ruler with a ploidy of three was applied as scaling mode with a total cellular protein concentration of 200 g/liter. Quantification accuracy was further estimated using the standard plugin parameters.

### Mathematical modeling

Implementation of all mathematical models was performed in the form of ODEs in MATLAB 2017a. Parameter optimization and un-certainty analysis was performed using the data2dynamics framework for MATLAB (Raue et al, 2015) and model analysis was performed using the ode15s solver in MATLAB. Detailed information about the mathematical model and its underlying assumptions is given in Supplemental Data 1; the MATLAB code of the core RIG-I model is available as Supplemental Data 2.

### Parameter estimation of the RIG-I core model

Total intracellular protein levels were gathered from a deep pro-teomic analysis of A549 and HepG2 cells (Supplemental Data 1). 5′ppp-dsRNA was assumed to be uniformly distributed post–electro-transfection, corresponding to a cytoplasmic dsRNA con-centration of 2 nM. No basal RIG-I signaling was accounted for, thus, yielding all phosphorylated protein levels to be zero before stimulation. In addition, in the absence of any stimulus, no free IκBα was considered. Total protein levels were converted to concen-trations using the reported cytoplasmic and nuclear volume of A549 cells (Vcyt = 1.2 × 10$^{-12}$ L and Vnucl = 4.7 × 10$^{-13}$ L [Jiang et al, 2010]), Nine kinetic rate constants were derived from the literature, whereas the remaining rate constants were optimized during the fitting process. The cytoplasmic degradation rate of 5′ppp-dsRNA was set to the fitted intracellular degradation rate constants of HCV RNA after electro-transfection (Binder et al, 2013). The reported half-lives for RIG-I proteins in HepG2 cells were used to define the basal RIG-I degradation rate constant and the basal RIG-I synthesis rate constant was derived by utilizing $k_{syn}$ = [RIGI]$_{t=0}$·$\mu_{RIG-I}$ (Arimoto et al, 2007). Degradation of IκBα and the binding rate constant to NF-κB by IκBα were taken from the work of Hoffmann et al (2002), expecting no strong differences in their reaction kinetics between A549 and human Jurkat cell lines. Because of the lack of experi-mental data regarding the temporal RIG-I activation and MAVS complex formation dynamics, reported values for k$_{RIG-I}$, k$_{MAVS,}$ and b$_{RIG-I}$ were employed from an existing mathematical model of the virus-triggered type I IFN-signaling pathway (Zou et al, 2010). The degradation rate constant of IFN-$\beta$ mRNA was further derived from its reported half-life in immortalized human bronchial epithelial cells (Abe et al, 2012). The 11 remaining rate constants were opti-mized using experimentally observed Western blot intensities and

IFN-$\beta$ qRT-PCR data. Therefore, experimentally observed, normalized intensity values were linked to the concentrations of model species with the help of scaling factors and background intensities (Supplemental Data 1). Background intensities for phosphorylated proteins were defined as mean of normalized intensity values for the first two time points of the short-period experiment (0 min, 1 min). The remaining scaling factors were optimized during the optimization process alongside the kinetic rate constants. Parameter fitting was performed by simultaneously minimizing the negative logarithm of the likelihood for the observed experimental data given our model parameters using a multi-start optimization method based on latin hypercube sampling (LHS) as implemented in the data2dynamics framework for MATLAB (Raue et al, 2013, 2015). The parameters were log10-transformed before optimization and a search space of at least six orders of magnitude was applied. To ensure convergence towards a global minimum in the multi-dimensional parameter space, 1,500 runs were performed using LHS-sampled initial parameter guesses. The sorted final goodness of fit for every optimization run is shown in Supplemental Data 1 and shows repeated convergence towards a global optimum.

### Parameter estimation of the coupled model of antiviral signaling

Whenever possible, parameter values were taken from the previously established RIG-I core model or the original IFN pathway model (Maiwald et al, 2010). Already existing protein and mRNA species of the core model constitute the same initial values as reported previously and total protein levels of IFN pathway components were adjusted based on proteomics data. Basal levels of JAK/STAT model species were determined by simulating their steady-state concentrations in the absence of any stimulus (Supplemental Data 1). IFN and ISG levels were set to zero before stimulation.

To combine the RIG-I pathway model with the published JAK-STAT model, we included a production rate of IFNpre, that is, the stimulus of JAK-STAT signaling, by the IFN-$\beta$ mRNA species of our RIG-I pathway core model. The remaining structure of the JAK-STAT model was not changed, but merely additional ISG model species were introduced to better capture the antiviral state of a cell by our model. In total, we considered the expression of six genes in our model (CCL5, CXCL10, *IFIT1*, IFN-$\lambda$, RIG-I, MX1), in addition to the already existing up-regulated species I$\kappa$B$\alpha$, IFN$\beta$, IRF9, and SOCS. Expression of ISGs is under the regulation of phosphorylated IRF3 (*CCL5*, *CXCL10*, IFN-$\lambda$) or under the regulation of both phosphorylated IRF and active ISGF3 (*IFIT1*, *RIG-I*, *MX1*). Depending on the availability of qRT-PCR and Western blot data, ISGs are included at the transcript level (*CCL5*, *CXCL10*), the protein level (RIG-I) or at both levels (IFIT1, IFN-$\lambda$, MX1) as species in our model. Furthermore, we account for a first-order decay of mRNAs (denoted $\mu$) and, when a protein species is included in the model, a translation and protein degradation process.

Only a single rate constant ($k_{68}$) of the Maiwald model (Maiwald et al, 2010) was readjusted, thereby accounting for differences in the potential of secreted IFN to trigger JAK/STAT signaling in the experimental setups. All remaining reported rate constants were assumed to be identical in A549 and Huh7.5 cells. Among the newly introduced parameters, degradation rate constants were fixed based on reported half-lives whenever possible ($\mu_{CCL5,mRNA}$, $\mu_{CXCL10,mRNA}$, $\mu_{IFN-\lambda,mRNA}$, $\mu_{MX1,mRNA}$, $\mu_{IFIT1,mRNA}$, $\mu_{MX1}$, $\mu_{IFIT1}$) (Ronni et al, 1993; Lam et al, 2001; Marçais et al, 2006; Dhillon et al, 2007; Sharova et al, 2009; Schmid et al, 2015; Voigt & Yin, 2015; Schmidtke et al, 2019). The degradation of IFN in the supernatant was assumed to be negligible and set to 0. The remaining kinetic rate constants were optimized based on experimentally observed levels of RIG-I protein, IFIT1 protein, IRF9 protein, phosphorylated STAT2 protein, *IFIT1* mRNA, *MX1* mRNA, *IFNB1* mRNA levels, and IFN-$\alpha$ levels in the supernatant (Fig 5C). Model fitting was performed by minimizing the negative log likelihood using a multi-start approach based on LHS as implemented in the data2dynamics framework (Raue et al, 2013, 2015). Moreover, scaling factors and background intensities were introduced to link experimentally observed intensities to intracellular concentrations. The resulting goodness of fit for every optimization run is shown in Supplemental Data 1, highlighting convergence towards a global optimum.

### Uncertainty analysis

Structural and practical identifiability of the core and coupled model were assessed by calculating the profile likelihood estimates for every optimized parameter as implemented in the data2-dynamics framework (Raue et al, 2009, 2015). Briefly, starting from the optimal set of parameters, every parameter was stepwise-fixed at increasingly deviating values, whereas all other open parameters were readjusted. The relation between the goodness of fit for given changes in the parameter values allowed us to identify parameter regions and likelihood-based 95% confidence intervals (Raue et al, 2009). The resulting likelihood profiles for the core model are shown in Supplemental Data 1. The analysis revealed a linear dependency between the shared rate constant for the activation of TBK1/IKK$\varepsilon$ and the IRF3 phosphorylation rate constant by the two kinases. The shared rate constant for the activation of TBK1/IKK$\varepsilon$ was consequently fixed and the IRF3 phosphorylation rate was optimized. For all other fitted parameters of the RIG-I core model, no identifiability problems could be identified.

## Data Availability

The mass spectrometry data from this publication have been submitted to the PRIDE database and assigned the identifier PXD031406.

## Supplementary Information

## Acknowledgements

We want to thank Maike Drechsler for excellent technical assistance, the DKFZ research group Molecular Therapy of Virus-Associated Cancers for IncuCyte-related support, Christoph Stein-Thöringer, Nyssa Cullin, and

Rogier Gaiser for providing the MSD instrument and the associated technical support, Friedemann Weber and Georgios Panagiotidis (Giessen University) for experimental expertise, Konstantin Sparrer (Ulm University) for providing SARS-CoV2 ORF6, Volker Lohmann for the provision of cell lines, and Ralf Bartenschlager for providing an excellent research environment. We are grateful to the Helmholtz International Graduate School for Cancer Research for supporting SSB with a Ph.D. stipend. The work was supported by the following funders, who had no role in study design, data collection, data analysis or the decision to publish: BMBF e:Bio initiative ImmunoQuant (0316170C to M Binder, 0316170E to L Kaderali), DFG BI1693/1-2 (to M Binder), and project 272983813 (SFB/TRR179, TP11 to M Binder and A Pichlmair), ERASysApp SysVirDrug 031A602A to L Kaderali, DKFZ doctoral stipend to SS Burkart, an ERC consolidator Grant (ERC-CoG ProDAP, 817798), the Bavarian State Ministry of Science and Arts (Bavarian Research Network FOR-COVID), and the German Research Foundation (PI 1084/5, PI 1084/6, PI 1084/7 and TRR237/A07) to A Pichlmair. L Kaderali acknowledges funding from the Excellence Initiative of Mecklenburg-Vorpommern/European Social Fund (ESF) Grant ESF/14-BM-A55-0014/16, DFG Grant KA 2989/13-1 and the European Union, ViroInf Grant 955974.

## Author Contributions

SS Burkart: conceptualization, data curation, formal analysis, investigation, methodology, and writing—original draft, review, and editing.
D Schweinoch: data curation, formal analysis, investigation, visualization, methodology, and writing—original draft, review, and editing.
J Frankish: data curation, formal analysis, investigation, visualization, methodology, and writing—review and editing.
C Sparn: data curation and investigation.
S Wüst: validation, investigation, and methodology.
C Urban: investigation, visualization, and methodology.
M Merlo: investigation.
VG Magalhães: investigation, visualization, methodology, and writing—review and editing.
A Piras: investigation and methodology.
A Pichlmair: data curation, formal analysis, and funding acquisition.
J Willemsen: conceptualization, supervision, methodology, and writing—review and editing.
L Kaderali: conceptualization, data curation, formal analysis, supervision, funding acquisition, and writing—review and editing.
M Binder: conceptualization, formal analysis, supervision, funding acquisition, and writing—original draft, review, and editing.

## Conflict of Interest Statement

The authors declare that they have no conflict of interest.

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
