## [Reviewer comments · Life Science Alliance]

Life Science Alliance

High Resolution Kinetic Characterization of the RIG-I Signaling Pathway and the Antiviral Response

Sandy Burkart, Darius Schweinoch, Jamie Frankish, Carola Sparn, Sandra Wüst, Christian Urban, Marta Merlo, Vladimir Magalhães, Antonio Piras, Andreas Pichlmair, Joschka Willemsen, Lars Kaderali, and Marco Binder

DOI: <https://doi.org/10.26508/lsa.202302059>

Corresponding author(s): Marco Binder, German Cancer Research Center and Lars Kaderali, Greifswald University Hospital

Review Timeline:

Submission Date:	2023-03-24
Editorial Decision:	2023-04-27
Revision Received:	2023-06-23
Editorial Decision:	2023-07-14
Revision Received:	2023-07-24
Accepted:	2023-07-27

Scientific Editor: Novella Guidi

Transaction Report:

April 27, 2023

Re: Life Science Alliance manuscript #LSA-2023-02059

Dr. Marco Binder
German Cancer Research Center
Im Neuenheimer Feld 280
Heidelberg 69120
Germany

Dear Dr. Binder,

Thank you for submitting your manuscript entitled "High Resolution Kinetic Characterization and Modeling of RIG-I Signaling and the Antiviral Response" to Life Science Alliance. The manuscript was assessed by expert reviewers, whose comments are appended to this letter. We invite you to submit a revised manuscript addressing the Reviewer comments.

Thank you for this interesting contribution to Life Science Alliance. We are looking forward to receiving your revised manuscript.

Sincerely,

B. MANUSCRIPT ORGANIZATION AND FORMATTING:

Reviewer #1 (Comments to the Authors (Required)):

Review of Burkart, Binder_LSA_2023

In this work, the authors undertake the challenging task of carefully measuring and modeling the time courses of individual processes that take place immediately upon RIG-I stimulation, thereby providing the first precise view of different stages in the RLR signaling pathway. The experimental design is particularly impressive, with careful controls that enable meticulous dissection of each aspect of the pathway through time. This reviewer was particularly intrigued with the way the IFNAR KO line was used to investigate uncharacterized aspects of RLR-induced gene expression. Going far beyond monitoring of IRF3 translocation and IFN expression, the experiments in this paper are far more sophisticated than those conducted previously, enabling quantitative dissection of the sequential events before and after transcriptional activation. Perhaps most novel overall (Fig. 5), the authors provide the first insights into the kinetic behavior of genes that are expressed when RIG-I is activated as an ISG, rather than its initial activation as a PRR. This is important because it may shed light on a potential function for this secondary phase of RIG-I activation. Relevance of the work is further enhanced by studies of antiviral countermeasures, such as NS3/4A cleavage of MAVS, which has remained completely qualitative until now. That said, it seemed like the last section of the paper might be best presented in a separate manuscript (point 3, below). Overall, this is an impressive body of work that will advance the field of RLR signaling in productive direction.

1. Comment on Fig. 1, 3. The data presented in these figures is outstanding, enabling the investigators to definitively track dose-dependence and speed with great precision. However, actual kinetic rates shown here for stages of RIG-I activation (rapid rise before 1h and peak around 2h) are a relatively good match to those reported in Thoresen, despite the fact that the previous work employed an endosomal uptake approach with a liposomal formulation, and the present study employed the more direct electroporation. Why do the authors think this is the case? Could it be attributable to differences in the RNA stimulatory ligand used in the two studies? The previous work seems to suggest that endosomal uptake is not always slow and stochastic. It would be good to understand how the authors rationalize this.

2. In the opinion of this reviewer, the experiments described in Fig. 5 provide some of the most revealing new insights into the pathway, as they enable us to understand how IFN-induced RIG-I gene expression may contribute to the antiviral response. There is a tremendous amount of detailed description of individual expression trajectories, but the authors do not draw any clear conclusions about how these late-stage events might contribute to effective antiviral response. This reviewer felt that the lack of interpretation in this section of the paper represented a missed opportunity. It would help the community if the authors might wrap this section up more coherently, or devote some of the discussion to interpretation of the results.

3. This reviewer very much appreciated and data on the kinetic response to viral countermeasures, but this section of the paper seemed out of place. It would have been better to take this part out, and build a small, focused manuscript around the data in Fig. 7 and the last part of the paper.

Reviewer #2 (Comments to the Authors (Required)):

This excellent manuscript by Burkart et al. experimentally addresses a long-standing question in the field: what are the comprehensive kinetics and dynamics of RLR signaling upon dsRNA recognition? Although this question has been addressed in the past as a side-note in other publications, those other studies did not give this topic a comprehensive look, and/or relied on modeling without carefully crafted experimental data. Particularly exciting in this present study is the implementation of viral RLR antagonists in the experimental and modeling approach. This manuscript will be an important resource for the field.

Comments on each main point of the paper:

- RIG-I signaling upon dsRNA recognition is highly deterministic and synchronous. I agree with most conclusions of this section in the main text (regarding Figure 1), but am unclear about the choice of title. What is meant by deterministic? What is the signaling synchronous to?

Further, in this section, I suggest replacing the word "modality" in line 154 with the word "synchronized timing". The word "modality" to me implies lipids vs electroporation per se, which are not comparable because of their difference in dsRNA uptake

dynamics.

Additionally, based on the materials and methods section, uptake by electroporation is achieved during the 10 ms pulse. Then, excess dsRNA is washed off, limiting the timing of dsRNA uptake to a few minutes at most. I was unable to find information on how long the lipofectamine 2000 complexes were left on the cells ("following manufacturer's protocol"), however, this information is critical to interpret the data in Figure 1. Please add this to the materials section and expand on the timing aspect in the section related to Figure 1.

Finally, the authors should comment here on the potential confounder of overexpressing IRF3 at baseline. Although this is addressed later in experiments with the ROSA26-IRF3, it is important to state the limitation upfront.

- Synchronous RIG-I stimulation results in a fast onset of RLR pathway signaling. Agree with design and interpretation of experiments. For the title, the word "fast" should be replaced by a more specific word choice.
- Dynamic RIG-I signaling model accurately reproduces activation kinetics of essential pathway components. Agree with design and interpretation of experiments.
- IFN feedback upon RLR stimulation is essential for high and sustained ISG expression, but not for IFN production. Agree with design and interpretation of experiments.
- Expanded RIG-I pathway model comprising IFN feedback accurately reproduces the full cell-intrinsic antiviral response. Agree with design and interpretation of experiments.
- RIG-I signaling dynamics can be accurately modeled in different cell lines. Agree with design and interpretation of experiments.
- The impact of viral antagonists on antiviral signaling dynamics can be properly simulated by the established mathematical model. Agree with design and interpretation of experiments. I suggest highlighting the SARS-CoV-2 ORF6 story more, as the findings could provide context for the existing controversial literature on ORF6 phenotypes (or lack thereof).

Additional issues to be addressed:

- I really enjoyed reviewing this manuscript, but suggest shortening the text considerably. For example, the building of the model does not need to be described in the main text as extensively (unless the authors feel the model building itself is a major result). More specific language would help confer the same information in more straightforward ways. The discussion could use more structure, with more paragraphs, and leader and closer sentences for each paragraph to flush out the main takeaways.
- Consider revising word choices such as "picked", "notoriously", "a couple of", and others.
- I strongly suggest changing color coding throughout the figures to reflect the different experimental setups and ease extraction of information for the reader relying on visuals.

Examples:

Figure 3, WT A549 is purple, IRF3 KO is turquoise and IRF3KO ROSA26-IRF3 is teal.

Same teal is used in Figure 4 for IRNR DKO, but this time WT is white.

Same teal is used in Figure 5 for ROSA26-IRF9, and this time WT is grey.

Same teal is used in Figure 6 for HepG2, with A549 in grey.

In Figure 7, HepG2 are in grey, like A549 in Figure 6, and HepG2 ORF6 in purple, like WT A549 in Figure 3.

Reviewer #3 (Comments to the Authors (Required)):

In the current manuscript, the authors demonstrate the kinetics of antiviral signaling through RIG-I using electroporation-based stimuli and mathematical simulation analysis. The authors show that electroporation of RIG-I substrate dsRNA into A549 cells can induce synchronized RIG-I signaling in a short period, and the mathematical modeling using the data obtained shows that it is sufficient to model the signaling dynamics. Interestingly, the effects of positive feedback regulation via IFNAR1/IFNLR were also examined using receptor-deficient A549 cells. Combined with the ODE model of IFN-stimulated cells, the model predicts gene expression regulation by RIG-I- and IFNR-mediated signaling for up to 24 hours. Moreover, the authors demonstrate that it can be used in the interaction between host immune signaling and viral antagonists, which may contribute to understanding the regulated induction of the IFN system in various viral infections.

Specific points,

1. As the authors point out, Bertolusso et al. (PLoS One, 2014) reported a similar modeling analysis of the RIG-I/TLR3 signaling pathway using dsRNA electroporation and A549 cells. In the current study, although the authors assert that "the RIG-I signaling pathway," but do not clearly explain the difference from the previous report except for the usage of 5'ppp-containing dsRNA as a RIG-I ligand. It needs to be clarified whether only RIG-I is exclusively involved in signaling in the system used in this work.
2. It is intriguing that signaling responses in different cell lines, such as A549 and HepG2, can be accurately predicted by adapting the initial protein concentrations in silico simulation. On the other hand, Bertolusso et al. observed the degradation of proteins, including RIG-I, after electroporation of dsRNA in A549 cells. But little consideration was given to the involvement of proteolysis or negative regulation in this work. Also, the RIG-I-mediated signaling pathway is known to be regulated by ubiquitination and/or deubiquitination. Discussion of the involvement of these factors may further shape the accuracy of the modeling in the current study.
3. The authors present the data showing cellular analysis and simulation models of four viral proteins that have been known to inhibit the IFN system at the induction or effector level. Interestingly, while the inhibitory effects of some antagonists are predicted in model simulations, the data suggest unknown protein functions. However, except for SARS-CoV-2 ORF6, the presented data focus on only the IFIT1 gene, and the data for the effect on IFN gene induction level should also be shown.

4. Discrepancies between the figure and the legend should be adjusted appropriately. For example, in Figs 3B and 5C, it is explained that "lines represent the average model fit and dashed lines represent confidence intervals." however, "line" and "dashed lines" are not clear. Also, in Supplemental Fig S5, the sample instructions in the figure and the text do not match.

Dear editor, dear reviewers.

First of all, we would like to thank you for the effort you put into evaluating our manuscript. We are really overwhelmed and touched by such positive feedback. Furthermore, we find all raised issues very thoughtful and constructive, and happily tried to accommodate them in our revised manuscript. Although it has not been raised as an explicit request by any of the three reviewers of this journal, we felt we could complement and solidify our study by including new experiments in primary human cells (human foreskin fibroblasts, new Supplementary Figure S6 and S7). Thereby, we believe, we can also more convincingly rebut / answer a few of the reviewers' comments and questions (see point-by-point response below). While not changing the overall conclusions and message of our initial manuscript, we hope the editor and reviewers agree these additional data make the manuscript even stronger, and the addition of two additional authors is justified (V.G.M. and M.M.).

We would like to thank all of you again and continue with our point-by-point reply below; please find our response in blue font.

Please note that the line numbering we refer to in this rebuttal letter relates to the file “Burkart et al., revised manuscript (complete), changes annotated” to make spotting changes easier; unfortunately, MS Word has a different line numbering when “track changes” is active.

Reviewer #1

In this work, the authors undertake the challenging task of carefully measuring and modeling the time courses of individual processes that take place immediately upon RIG-I stimulation, thereby providing the first precise view of different stages in the RLR signaling pathway. The experimental design is particularly impressive, with careful controls that enable meticulous dissection of each aspect of the pathway through time. This reviewer was particularly intrigued with the way the IFNAR KO line was used to investigate uncharacterized aspects of RLR-induced gene expression. Going far beyond monitoring of IRF3 translocation and IFN expression, the experiments in this paper are far more sophisticated than those conducted previously, enabling quantitative dissection of the sequential events before and after transcriptional activation. Perhaps most novel overall (Fig. 5), the authors provide the first insights into the kinetic behavior of genes that are expressed when RIG-I is activated as an ISG, rather than its initial activation as a PRR. This is important because it may shed light on a potential function for this secondary phase of RIG-I activation. Relevance of the work is further enhanced by studies of antiviral countermeasures, such as NS3/4A cleavage of MAVS, which has remained completely qualitative until now. That said, it seemed like the last section of the paper might be best presented in a separate manuscript (point 3, below). Overall, this is an impressive body of work that will advance the field of RLR signaling in productive direction.

As stated already above, we are extremely glad the reviewer appreciates our work as a significant advance of the field.

1. Comment on Fig. 1, 3. The data presented in these figures is outstanding, enabling the investigators to definitively track dose-dependence and speed with great precision. However, actual kinetic rates shown here for stages of RIG-I activation (rapid rise before 1h and peak around 2h) are a relatively good match to those reported in Thoresen, despite the fact that the previous work employed an endosomal uptake approach with a liposomal formulation, and the present study employed the more direct electroporation. Why do the authors think this is the case? Could it be attributable to differences in the RNA stimulatory ligand used in the two studies? The previous work seems to suggest that endosomal uptake is not always slow and stochastic. It would be good to understand how the authors rationalize this.

This is an excellent point and we happily discuss this now in our revised manuscript's discussion section, line 884ff (“Notably, also the study by Thoresen...”). In brief, the reviewer is right, the choice of RNA ligand (and particularly its high molarity) may play a role; while lipofection / endosomal uptake will always be slower, if high enough concentrations of RNA are transfected, absolute numbers of molecules entering the cells already early on will suffice to trigger strong RIG-I signaling. We rather see this as a mutual confirmation of Thoresen's and our findings, underscoring the previous underestimation of pathway kinetics. The experiments by Thoresen are also corroborated by our own, new experiments in Suppl. Fig. S7, where we use authentic virus (VSV) high titer spin-infection in primary cells and observe a comparably quick induction kinetics. This is included in the new discussion paragraph mentioned. See also comment 3 of reviewer #2.

2. In the opinion of this reviewer, the experiments described in Fig. 5 provide some of the most revealing new insights into the pathway, as they enable us to understand how IFN-induced RIG-I gene expression may contribute to the antiviral response. There is a tremendous amount of detailed description of individual expression trajectories, but the authors do not draw any clear conclusions about how these late-stage events might contribute to effective antiviral

response. This reviewer felt that the lack of interpretation in this section of the paper represented a missed opportunity. It would help the community if the authors might wrap this section up more coherently, or devote some of the discussion to interpretation of the results.

Again, we fully agree with the reviewer: possibly the most important but also most complex aspect of antiviral signaling is its intricate feedback and feedforward regulation. We intentionally focused on describing individual trajectories and not interpreting or suggesting too much (even if it were very educated guesses). However, we now included a sentence to the end of the paragraph describing Fig. 4, line 451ff. We furthermore re-structured the discussion as per reviewer #2's request and to underscore the high relevance of the topic as the reviewer #1 rightfully pointed out, we now included a separate topic "Relevance of negative and positive feedback and feedforward regulation" (lines 1061ff).

3. This reviewer very much appreciated and data on the kinetic response to viral countermeasures, but this section of the paper seemed out of place. It would have been better to take this part out, and build a small, focused manuscript around the data in Fig. 7 and the last part of the paper.

We do understand the reviewer's comment and also appreciate their suggestion to "out-source" the viral counteraction part into a separate manuscript. However, we think of this part as an important example for research questions that benefit tremendously from studying the dynamics of processes rather than single endpoints. Also, reviewers #2 and #3 were more positive about this part and encouraged us to even include some more data, in particular to SARS-CoV-2 ORF6, which we did (line 773ff, 1244ff, Suppl. Fig. S8). Lastly, we now include new experiments in primary cells using virus infection in the end of our "signaling dynamics / modelling" part (line 592ff), which we feel makes a smoother transition to the viral countermeasures part now. We hope this reviewer find our decision acceptable.

Reviewer #2

This excellent manuscript by Burkart et al. experimentally addresses a long-standing question in the field: what are the comprehensive kinetics and dynamics of RLR signaling upon dsRNA recognition? Although this question has been addressed in the past as a side-note in other publications, those other studies did not give this topic a comprehensive look, and/or relied on modeling without carefully crafted experimental data. Particularly exciting in this present study is the implementation of viral RLR antagonists in the experimental and modeling approach. This manuscript will be an important resource for the field.

As stated above, we are extremely happy to read the reviewer's so positive feedback to our work and his assessment of it being important to the field.

- RIG-I signaling upon dsRNA recognition is highly deterministic and synchronous. I agree with most conclusions of this section in the main text (regarding Figure 1), but am unclear about the choice of title. What is meant by deterministic? What is the signaling synchronous to?

We appreciate this might not have been made clear enough. We have now removed "synchronous" from the title and explained both terms (or our use of them) more precisely in the text (e.g. line 159ff, 204, 211f). We further replaced "synchronous" by "simultaneous" throughout the manuscript where we saw fit.

Further, in this section, I suggest replacing the word "modality" in line 154 with the word "synchronized timing". The word "modality" to me implies lipids vs electroporation per se, which are not comparable because of their difference in dsRNA uptake dynamics.

We thank the reviewer for this comment— we were not making this clear enough. In fact, we intended to say exactly what the reviewer states, and now rephrased the sentence in line 220 to specify "RNA-uptake kinetics of the chosen transfection modality" determine the synchronicity of the onset of signaling. We hope this has resolved the confusion.

Additionally, based on the materials and methods section, uptake by electroporation is achieved during the 10 ms pulse. Then, excess dsRNA is washed off, limiting the timing of dsRNA uptake to a few minutes at most. I was unable to find information on how long the lipofectamine 2000 complexes were left on the cells ("following manufacturer's protocol"), however, this information is critical to interpret the data in Figure 1. Please add this to the materials section and expand on the timing aspect in the section related to Figure 1.

This is an important point, and the reviewer is absolutely right in spotting that we have not made this clear before. We now explicitly state this in the main text (line 156ff) and additionally include a statement in the methods section

("lipoplexes were left on cells through the duration of the experiment", line 1333). On somewhat related terms, see also our response to comment 1 by reviewer #1.

Finally, the authors should comment here on the potential confounder of overexpressing IRF3 at baseline. Although this is addressed later in experiments with the ROSA26-IRF3, it is important to state the limitation upfront.

Once more, this is an excellent point. We are uncertain whether the reviewer assumed we would use IRF3 overexpressing cells all along in the manuscript? This is not the case, Figure 1 is the only experiment where we used them. Nonetheless, we agree for this figure the issue is relevant and we now address this point in the main text, line 204ff and in the discussion, line 879ff.

For this reviewer, we would like to show the below set of data, in which we (years back) assessed kinetics of phosphorylation of eGFP-IRF3 and endogenous IRF3 in the eGFP-IRF3 overexpressing cells, comparing it to IRF3 phosphorylation in wildtype cells. One can appreciate more stable phosphorylation of IRF3-GFP over time, however, the kinetics of activation in the first hour is very comparable.

- Synchronous RIG-I stimulation results in a fast onset of RLR pathway signaling. Agree with design and interpretation of experiments. For the title, the word "fast" should be replaced by a more specific word choice.

We replaced "fast" by "immediate".

- The impact of viral antagonists on antiviral signaling dynamics can be properly simulated by the established mathematical model. Agree with design and interpretation of experiments. I suggest highlighting the SARS-CoV-2 ORF6 story more, as the findings could provide context for the existing controversial literature on ORF6 phenotypes (or lack thereof).

We thank the reviewer for appreciating this part of our manuscript. Following reviewer #3's request to include IFNB1 mRNA in addition to IFIT1 for NS3/4A, Npro and NS5, we now provide these data in Suppl. Fig. S8. For NS5, this does not fully mirror our IFIT1 findings, so we now toned down our interpretation a bit. On the other hand, following this reviewer's suggestion, we have now included another set of data on ORF6, namely IFIT1-promoter luciferase reporter assays in both, A549 and HepG2 cells. While we are able to see an impact of ORF6 on IFIT1 and IFNB1 mRNA induction only in HepG2, we can see a significant effect on IFIT1-promoter luciferase in both cell lines. As ORF6 was reported to affect transcription factor nuclear import *AND* nuclear export of mature ISG mRNA, we speculate our findings might reflect a stronger impact on mRNA export (and, hence, indirectly also luciferase translation) in particular in A549 cells. We discuss this now somewhat more in detail (line 773ff, lines 1114-1249 in the discussion).

Additional issues to be addressed:

- I really enjoyed reviewing this manuscript, but suggest shortening the text considerably. For example, the building of the model does not need to be described in the main text as extensively (unless the authors feel the model building itself is a major result). More specific language would help confer the same information in more straightforward ways. The discussion could use more structure, with more paragraphs, and leader and closer sentences for each paragraph to flush out the main takeaways.

We really tried hard to make the text more concise and straightforward. We have removed unnecessary attributes and adverbs, literature-review like passages, and other extraneous text throughout the manuscript (see change tracked version of manuscript file). We further re-structured the discussion as the reviewer suggested, which we actually like a

lot; we believe it now is much more approachable for the reader. Lastly, we did shorten the modelling part of the manuscript as much as we could, however, we *DO* consider it a central part of the story. After all, it took a lot of time and effort to establish (PhD thesis of D. Schweinoch) and will– to the more bio-computational part of the readership– hopefully be a valuable basis for further developments.

We want to be honest here: we miserably failed in “considerably” shortening the manuscript. We fear that for every word we shortened it by, we inserted another one at a different position, mostly to fulfil the requests and suggestions by the reviewers. We nonetheless hope the reviewer agrees in that, still, the quality and readability of our manuscript has significantly improved by careful revision of the text.

- Consider revising word choices such as "picked", "notoriously", "a couple of", and others.

As part of the textual revisions described in our reply above, we tried our best to spot all occurrences of these and similar words and replaced them by more neutral and professional language.

- I strongly suggest changing color coding throughout the figures to reflect the different experimental setups and ease extraction of information for the reader relying on visuals. Examples: Figure 3, WT A549 is purple, IRF3 KO is turquoise and IRF3KO ROSA26-IRF3 is teal. Same teal is used in Figure 4 for IRNR DKO, but this time WT is white. Same teal is used in Figure 5 for ROSA26-IRF9, and this time WT is grey. Same teal is used in Figure 6 for HepG2, with A549 in grey. In Figure 7, HepG2 are in grey, like A549 in Figure 6, and HepG2 ORF6 in purple, like WT A549 in Figure 3.

We have carefully gone through our figures and reconsidered our choice of colors, but we cannot fully agree with the reviewer that finding a separate color (that should still be easily discernable all others) for each and every cell line and condition would be possible and desirable. We did, however, make several adjustments and now always use gray for the reference / control sample (no matter the actual cell line) and teal as the primary color for all experimental samples. In Fig. 7, when we need to color-code an increasing quantity (antagonists), we used a gradient from purple to blue. We hope this scheme is now more coherent and less confusing.

Reviewer #3

In the current manuscript, the authors demonstrate the kinetics of antiviral signaling through RIG-I using electroporation-based stimuli and mathematical simulation analysis. The authors show that electroporation of RIG-I substrate dsRNA into A549 cells can induce synchronized RIG-I signaling in a short period, and the mathematical modeling using the data obtained shows that it is sufficient to model the signaling dynamics. Interestingly, the effects of positive feedback regulation via IFNAR1/IFNLR were also examined using receptor-deficient A549 cells. Combined with the ODE model of IFN-stimulated cells, the model predicts gene expression regulation by RIG-I- and IFNR-mediated signaling for up to 24 hours. Moreover, the authors demonstrate that it can be used in the interaction between host immune signaling and viral antagonists, which may contribute to understanding the regulated induction of the IFN system in various viral infections.

We thank the reviewer for their assessment and accurate summary of our manuscript, in particular their appreciation of our IFNAR1/IFNLR double-KO, which we believe is a first in all RLR literature and important for our understanding of primary *versus* secondary effects in the antiviral response.

Specific points

1. As the authors point out, Bertolusso et al. (PLoS One, 2014) reported a similar modeling analysis of the RIG-I/TLR3 signaling pathway using dsRNA electroporation and A549 cells. In the current study, although the authors assert that "the RIG-I signaling pathway," but do not clearly explain the difference from the previous report except for the usage of 5'ppp-containing dsRNA as a RIG-I ligand. It needs to be clarified whether only RIG-I is exclusively involved in signaling in the system used in this work.

We appreciate this reviewer has a thorough background with mathematical models, including those of the RLR system. Initially, we decided to not discuss previous literature on RIG-I models too much in order to focus on the biological ("wet-lab") readers. However, also given our own argument (to reviewer #2 above) that we consider the modeling part important and central to the manuscript, we absolutely agree with the reviewer that we should strengthen our discussion particularly of the Bertolusso paper, which is the strongest of all previous models. We now more extensively mention it in the discussion, see below here and below point 2 in the following.

Regarding the concrete question of the RIG-I specificity of the chosen ligand, we have assessed this question amply in prior work, e.g. in Wüst et al. and Willemsen et al. (ref 60 and 61) by RIG-I KO (and IRF3-KO to rule out involvement

of IRF3-independent activities by TLR3 as proposed by Bertolusso). We further found no indication that TLR3 is notably expressed or stimulation would occur to any relevant extent in A549 cells (Plociennikowska et al., ref 62). We discuss this now in lines 891ff, 935ff and 982ff.

2. It is intriguing that signaling responses in different cell lines, such as A549 and HepG2, can be accurately predicted by adapting the initial protein concentrations in silico simulation. On the other hand, Bertolusso et al. observed the degradation of proteins, including RIG-I, after electroporation of dsRNA in A549 cells. But little consideration was given to the involvement of proteolysis or negative regulation in this work. Also, the RIG-I-mediated signaling pathway is known to be regulated by ubiquitination and/or deubiquitination. Discussion of the involvement of these factors may further shape the accuracy of the modeling in the current study.

We thank the reviewer for appreciating this stunning conservation of the pathway across different cell types / lines. We now even widened the scope of cells and stimulation systems to primary cells (human primary foreskin fibroblasts, HFF) in both, dsRNA electro-transfection and also authentic virus (VSV). We did not assess protein levels in HFF and have not attempted to fit our model to this cell type, but find the qualitative kinetics / dynamics of pathway activation are amazingly similar to what we describe for A549 and HepG2. We think this further strengthens our point that the pathway is highly deterministic and functions similarly across a large variety of cell types. These new experiments can be found in Suppl. Fig. 6 and 7, described in the paragraph starting from line 592, and discussed e.g. at line 1045ff.

The second point raised here by the reviewer is very good and highly important, too: negative regulation of the pathway by protein degradation, ubiquitylation or other regulatory events. We have recognized that we have so far explained and discussed this inadequately (basically not at all), and apologize for this lapse. Of course, we are aware of negative regulation of the pathway (e.g. see our paper on neg. regulation of RIG-I by phosphorylation, ref. 61). We now include explicit discussion of this in our manuscript, e.g. line 985ff specifically on protein degradation but also further negative regulation; we also added a full paragraph on discussing the role of negative and positive feedback, as well as positive feedforward (as also suggested by reviewer #1) starting from line 1061. In brief, we neglected negative feedback regulation in our model, as it turned out not to be required for accurately simulating signaling under our experimental framework. This most likely largely owes to the fact that our RIG-I stimulus is very transient (pulse transfection) and, hence, delayed negative regulation of RIG-I and its downstream signaling will remain ineffective under these circumstances.

3. The authors present the data showing cellular analysis and simulation models of four viral proteins that have been known to inhibit the IFN system at the induction or effector level. Interestingly, while the inhibitory effects of some antagonists are predicted in model simulations, the data suggest unknown protein functions. However, except for SARS-CoV-2 ORF6, the presented data focus on only the IFIT1 gene, and the data for the effect on IFN gene induction level should also be shown.

We appreciate the reviewer finds our analyses regarding viral counteraction interesting. The reviewer raises a very valid point that we have now addressed in the revised manuscript. We now measured IFNB1 mRNA in all samples and provide these results in the new Suppl. Fig. S8. While for NS3/4A and Npro IFNB1 behaved as expected and largely comparable to IFIT1, it did not replicate the effect of Dengue NS5 on early gene induction. We have therefore toned down our (anyways speculative) hypothesis that NS5 might have additional activities in the RLR pathways, e.g. in lines 732ff in the main text, and also in the discussion starting on line 1102 (specifically 1107ff).

4. Discrepancies between the figure and the legend should be adjusted appropriately. For example, in Figs 3B and 5C, it is explained that "lines represent the average model fit and dashed lines represent confidence intervals." however, "line" and "dashed lines" are not clear. Also, in Supplemental Fig S5, the sample instructions in the figure and the text do not match.

We thank the reviewer for point out these discrepancies! We corrected the figure legends for Fig. 3B, 5C and Suppl. Fig. S5.

July 14, 2023

RE: Life Science Alliance Manuscript #LSA-2023-02059R

Dr. Marco Binder
German Cancer Research Center
Im Neuenheimer Feld 280
Heidelberg 69120
Germany

Dear Dr. Binder,

Thank you for submitting your revised manuscript entitled "High Resolution Kinetic Characterization and Modeling of RIG-I Signaling and the Antiviral Response". We would be happy to publish your paper in Life Science Alliance pending final revisions necessary to meet our formatting guidelines.

- please upload your main manuscript text as an editable doc file
- please upload your Tables in editable .doc or Excel format
- Tables should be numbered consecutively with Arabic numerals (1, 2, 3, 4)
- please upload all figure files as individual ones, including the supplementary figure files; all figure legends should only appear in the main manuscript file
- please add the Abstract and a Summary Blurb/Alternate Abstract to our system
- please add the Twitter handle of your host institute/organization as well as your own or/and one of the authors in our system
- please note that titles in the system and on the manuscript file must match
- please exclude figures from the manuscript text and upload them separately
- please add your main, supplementary figure, and table legends to the main manuscript text after the references section
- please consult our manuscript preparation guidelines <https://www.life-science-alliance.org/manuscript-prep> and make sure your manuscript sections are in the correct order
- please be sure to insert all Authors in the Authors Contribution section
- please add callouts for Figures S1A-B, S4A-E, S6A-D, S7A-E, S8D, E to your main manuscript text
- for the Modelling Supplement file: please remove the chapter 1 marking, considering there is no chapter 2. Please remove the References and incorporate them into the main Reference list instead. Please provide a word file as well along with the PDF file.

Figure checks:

- please add scale bars for Figure 1A

A. FINAL FILES:

- An editable version of the final text (.DOC or .DOCX) is needed for copyediting (no PDFs).
- High-resolution figure, supplementary figure and video files uploaded as individual files: See our detailed guidelines for

preparing your production-ready images, <https://www.life-science-alliance.org/authors>

B. MANUSCRIPT ORGANIZATION AND FORMATTING:

Sincerely,

Reviewer #1 (Comments to the Authors (Required)):

The manuscript is now suitable for publication

Reviewer #2 (Comments to the Authors (Required)):

This excellent manuscript by Burkart et al. experimentally addresses a long-standing question in the field: what are the comprehensive kinetics and dynamics of RLR signaling upon dsRNA recognition? Although this question has been addressed in the past as a side-note in other publications, those other studies did not give this topic a comprehensive look, and/or relied on modeling without carefully crafted experimental data. Particularly exciting in this present study is the implementation of viral RLR antagonists in the experimental and modeling approach. This manuscript will be an important resource for the field. All my previous concerns have been satisfactorily addressed.

Reviewer #3 (Comments to the Authors (Required)):

In the revised manuscript, the authors have adequately addressed the issues pointed out by the reviewer. The current findings will provide valuable advances in understanding RLR-mediated signaling and its interaction with viral antagonists.

Point-by-point reply to the editors raised issues:

-please upload your main manuscript text as an editable doc file

Done

-please upload your Tables in editable .doc or Excel format

Done

-Tables should be numbered consecutively with Arabic numerals (1, 2, 3, 4)

Done

-please upload all figure files as individual ones, including the supplementary figure files; all figure legends should only appear in the main manuscript file

Done

-please add the Abstract and a Summary Blurb/Alternate Abstract to our system

Done

-please add the Twitter handle of your host institute/organization as well as your own or/and one of the authors in our system

Done

-please note that titles in the system and on the manuscript file must match

checked

-please exclude figures from the manuscript text and upload them separately

Done

-please add your main, supplementary figure, and table legends to the main manuscript text after the references section

Done

-please consult our manuscript preparation guidelines <https://www.life-science-alliance.org/manuscript-prep> and make sure your manuscript sections are in the correct order

Done

-please be sure to insert all Authors in the Authors Contribution section

Done

-please add callouts for Figures S1A-B, S4A-E, S6A-D, S7A-E, S8D, E to your main manuscript text

Done

-for the Modelling Supplement file: please remove the chapter 1 marking, considering there is no chapter 2. Please remove the References and incorporate them into the main Reference list instead. Please provide a word file as well along with the PDF file.

Done

Figure checks:

-please add scale bars for Figure 1A

Done

July 27, 2023

RE: Life Science Alliance Manuscript #LSA-2023-02059RR

Dr. Marco Binder
German Cancer Research Center
Im Neuenheimer Feld 280
Heidelberg 69120
Germany

Dear Dr. Binder,

Thank you for submitting your Research Article entitled "High Resolution Kinetic Characterization of the RIG-I Signaling Pathway and the Antiviral Response". It is a pleasure to let you know that your manuscript is now accepted for publication in Life Science Alliance. Congratulations on this interesting work.

DISTRIBUTION OF MATERIALS:

Again, congratulations on a very nice paper. I hope you found the review process to be constructive and are pleased with how the manuscript was handled editorially. We look forward to future exciting submissions from your lab.

Sincerely,
